# Learning to Incentivize in Repeated Principal-Agent Problems with Adversarial Agent Arrivals

Junyan Liu [* 1]   Arnab Maiti [* 1]   Artin Tajdini [* 1]   Kevin Jamieson [1]   Lillian J. Ratliff [2]

## Abstract

We initiate the study of a repeated principal-agent problem over a finite horizon $T$, where a principal sequentially interacts with $K \geq 2$ types of agents arriving in an *adversarial* order. At each round, the principal strategically chooses one of the $N$ arms to incentivize for an arriving agent of *unknown type*. The agent then chooses an arm based on its own utility and the provided incentive, and the principal receives a corresponding reward. The objective is to minimize regret against the best incentive in hindsight. Without prior knowledge of agent behavior, we show that the problem becomes intractable, leading to linear regret. We analyze two key settings where sublinear regret is achievable. In the first setting, the principal knows the arm each agent type would select greedily for any given incentive. Under this setting, we propose an algorithm that achieves a regret bound of $\mathcal{O}(\min\{\sqrt{KT \log N}, K\sqrt{T}\})$ and provide a matching lower bound up to a $\log K$ factor. In the second setting, an agent's response varies smoothly with the incentive and is governed by a Lipschitz constant $L \geq 1$. Under this setting, we show that there is an algorithm with a regret bound of $\widetilde{\mathcal{O}}((LN)^{1/3}T^{2/3})$ and establish a matching lower bound up to logarithmic factors. Finally, we extend our algorithmic results for both settings by allowing the principal to incentivize multiple arms simultaneously in each round.

---

[*]Equal contribution   [1]Paul G. Allen School of Computer Science & Engineering, University of Washington, Seattle, USA [2]Department of Electrical and Computer Engineering, University of Washington, Seattle, USA. Correspondence to: Junyan Liu <junyanl1@cs.washington.edu>, Arnab Maiti <arnabm2@cs.washington.edu>, Artin Tajdini <artin@cs.washington.edu>.

*Proceedings of the 42^{nd} International Conference on Machine Learning*, Vancouver, Canada. PMLR 267, 2025. Copyright 2025 by the author(s).

## 1. Introduction

The repeated principal-agent model captures sequential decision-making scenarios where a principal strategically incentivizes agents over multiple rounds to act toward a long-term objective. This framework is central to many real-world applications, such as online platforms offering discounts to influence purchasing behavior, insurance companies designing contracts without full knowledge of customers' risk levels, and crowdsourcing platforms structuring payments to ensure high-quality contributions. In these settings, the principal cannot directly control agents' actions but influence them by proposing incentives, often under asymmetric information. Typically, asymmetric information introduces two key challenges, including *moral hazard* (Bolton & Dewatripont, 2004; Ho et al., 2014; Kaynar & Siddiq, 2023) and *adverse selection* (Scheid et al., 2024b; Dogan et al., 2023b;a). In moral hazard settings, the agents' actions cannot be directly observed by the principal, requiring incentive design to encourage agents to take actions in favor of the principal's interest. A common example is crowdsourcing where a task requester (principal) cannot directly observe the effort level of workers (agents). Adverse selection, on the other hand, arises when the agents' types or preferences are unknown. For example, in insurance, the insurer (principal) may not know whether a customer (agent) is high-risk, complicating the contract design.

While repeated principal-agent problems have been extensively studied, prior works heavily focus on the scenarios where the principal repeatedly contracts with a fixed and unknown type (Scheid et al., 2024b; Dogan et al., 2023a;b) or a random type of agent drawn stochastically from *a fixed underlying distribution* (Ho et al., 2014; Gayle & Miller, 2015). However, these two assumptions may not hold in many real-world applications. For example, in online shopping, discount ads displayed on a website are visible to all users, and the sequence of users who respond may not follow a stochastic pattern. Another example is crowdsourcing, where the sequence of crowdworkers who respond to a task could be arbitrary and subject to herding behavior through online forums (Horton et al., 2011). To capture the uncertainty of agent arrivals, we initiate the study of the repeated principal-agent problem with adversarial agent arrivals. To

our knowledge, this is the first study that examines this challenge in a repeated principal-agent framework.

Apart from agent arrival patterns, the agent response also plays a critical role in incentive design. Consider online shopping platforms where customers only make a purchase when a discount reaches a historical low or falls below their personal threshold. Similarly, crowdworkers in online labor markets often accept tasks only if the payout exceeds a minimum expectation. Such behaviors suggest a decision-making process governed by strict thresholds. To model this, we start by considering the *greedy response model* widely studied in the literature (Ho et al., 2014; Zhu et al., 2022; Dogan et al., 2023b; Scheid et al., 2024b; Ben-Porat et al., 2024), where an agent chooses an action that maximizes their utility, i.e., preference plus incentive. In this model, the agent plays an arm only when the incentive on this arm crosses a predefined level.

Ideally, we seek an approach that adapts to adversarial agent arrivals without prior knowledge of agent types, that is, how incentives influence the threshold values driving their decisions. Unfortunately, we will show that this is intractable, in the sense that no algorithm can compete with the single-best incentive in hindsight without that prior knowledge. Indeed, our lower bound exploits the fact that even a tiny change in incentives can lead to a drastic shift in agent decisions, and consequently, the principal's reward. This non-smooth sensitivity to incentives defies intuition and suggests the need for an alternative model where agent decisions respond more gradually to incentives. For instance, when purchasing daily necessities, a small change in discounts should not drastically alter a customer's decision but instead gradually influence their likelihood of buying. This distinction motivates the need for a more flexible response model.

Moreover, the smooth decision model can be viewed as an extension of the greedy model by allowing agents to incorporate smooth noise into their preferences. For example, if an agent adds i.i.d. Gumbel noise $\eta^t \sim \text{Gumble}(\mathbf{0}, 1)^N$ to their preference vector $\mu^{j_t}$, the resulting choice model becomes the Logit discrete choice model which is prevalent in economics literature (Train, 2009); and if an agent adds i.i.d. Gaussian noise $\eta^t \sim \mathcal{N}(\mathbf{0}, I)$ to their preference vector $\mu^{j_t}$, the resulting choice model becomes Lipschitz (see Appendix E.2).

### 1.1. Problem Statement

We study the repeated principal-agent problem, where a principal incentivizes $K$ types of agents to choose arms from the set $[N] = \{1, \ldots, N\}$. The principal has a *known* reward vector $v = (v_1, \ldots, v_N) \in [0,1]^N$. Each agent type $j \in [K]$ has an associated preference vector $\mu^j = (\mu_1^j, \ldots, \mu_N^j) \in [0,1]^N$.

The interaction unfolds over a finite horizon $t = 1, \ldots, T$, where agents arrive in an adversarial order. In each round $t$, an agent of type $j_t \in [K]$ arrives (where $j_t$ is unobserved by principal), and the principal simultaneously chooses an incentive vector $\pi_t = (\pi_{t,1}, \ldots, \pi_{t,N}) \in \mathcal{D}$, where $\mathcal{D} \subseteq [0,1]^N$ denotes the principal's decision space. The agent then selects an arm $a(\pi_t, j_t)$, taking $\pi_t$ and $j_t$ into account. The principal observes only the chosen arm $a(\pi_t, j_t)$ and receives the following utility: $U(\pi_t, j_t) = v_{a(\pi_t, j_t)} - \pi_{t, a(\pi_t, j_t)}$.

The objective of principal is to minimize the regret:

$$R_T = \sup_{\pi \in \mathcal{D}} \mathbb{E}\left[\sum_{t=1}^{T} \left(U(\pi, j_t) - U(\pi_t, j_t)\right)\right].$$

**Incentive structures.** We consider two types of incentives: single-arm incentives and general incentives.

*Single-arm incentive*: the principal can incentivize at most one arm, meaning the incentive vector has at most one nonzero coordinate. The decision space is

$$\mathcal{D} = \{x \in [0,1]^N : |\text{supp}(x)| \leq 1\}.$$

*General incentive*: the principal can incentivize multiple arms, meaning the principal can choose any incentive vector in $[0,1]^N$. The decision space is $\mathcal{D} = [0,1]^N$.

**Arm Selection Models** We analyze two models for agent decision-making, namely, the greedy choice model and the smooth choice model.

*Greedy choice model*: an agent of type $j_t$ observes the incentive vector $\pi_t$ and best responds by deterministically choosing the arm

$$b(\pi_t, j_t) \in \arg\max_{i \in [N]} \left\{\mu_i^{j_t} + \pi_{t,i}\right\}.$$

Ties are broken arbitrarily but consistently across agent types. We assume that the principal knows the best response function $b(\cdot, \cdot)$, meaning the principal can determine $b(\pi, j)$ for any $\pi \in \mathcal{D}$ and $j \in [K]$.

*Smooth choice model*: the agent of type $j_t$ selects an arm $a(\pi_t, j_t)$ probabilistically based on an *unknown* distribution that varies smoothly with the incentive vector as follows.

**Assumption 1.1. (Smooth decision model)** For any two incentive vectors $\pi, \pi' \in \mathcal{D}$ and any agent type $j$, the probability distribution of arm selection satisfies:

$$\sum_{i=1}^{N} \left| \Pr[a(\pi, j) = i] - \Pr[a(\pi', j) = i] \right| \leq L \cdot ||\pi - \pi'||_\infty,$$

where $L \geq 1$ is a Lipschitz constant.

Table 1: Regret bounds under different agent behaviors and incentive models.

| Agent behavior | Incentive type | Upper bound | Lower bound |
|---|---|---|---|
| Unknown, Greedy choice | Single-arm | N/A | $\Omega(T)$ |
| Known, Greedy choice | Single-arm | $\tilde{\mathcal{O}}\left(\min\left\{\sqrt{KT\log(N)}, K\sqrt{T}\right\}\right)$ | $\tilde{\Omega}(\min\{\sqrt{KT\log(N)}, K\sqrt{T}\})$ |
| | General | $\mathcal{O}(K\sqrt{T\log(KT)})$ | – |
| Unknown, Smooth | Single-Arm | $\mathcal{O}\left(L^{1/3}N^{1/3}T^{2/3}\right)$ | $\Omega\left(L^{1/3}N^{1/3}T^{2/3}\right)$ |
| | General | $\mathcal{O}\left(L^{N/(N+2)}T^{(N+1)/(N+2)}\right)$ | – |

## 1.2. Contributions

All our results are summarized in Table 1. For the greedy response agent and single-arm incentive case, we show that any algorithm without prior knowledge of the agents' behaviors can be forced to incur linear regret. However, when the agent behaviors are known as a priori, that is we know the best response function of each agent but not which agent arrives at each round, we discretize the continuous incentive space and propose a reduction-based approach that turns the problem into an adversarial linear bandits problem with finite arms. The proposed algorithm achieves a $\tilde{\mathcal{O}}(\min\{\sqrt{KT\log(KN)}, K\sqrt{T}\})$ upper bound. Under this same setting of known types, we provide a $\tilde{\Omega}\left(\min\left\{K\sqrt{T}, \sqrt{KT\log(N)}\right\}\right)$ regret lower bound, implying that our upper bound is tight up to logarithmic factors.

For the case of greedy agent and known agents' behaviors, if the principal is allowed to incentive more than one arm simultaneously (also known as general incentives), we adopt the previous reduction to adversarial linear bandits, but more importantly, introduce a novel discretization, which enables our algorithm to achieve a $\tilde{\mathcal{O}}(K\sqrt{T})$[1] regret bound. The key idea behind the discretization is identifying a polytope for the large incentive space such that the extreme point of the polytope is close to the optimal incentive.

Finally, for the smooth response agent and single-arm incentive case, we propose an algorithm which achieves a $\widetilde{\mathcal{O}}(L^{1/3}N^{1/3}T^{2/3})$ regret bound where $L \geq 1$ is a known Lipschitz constant. We further show the tightness of this upper bound as it matches our $\Omega(L^{1/3}N^{1/3}T^{2/3})$ lower bound up to logarithmic factors. We also show that for the general incentive case, $\widetilde{\mathcal{O}}(L^{N/(N+2)}T^{(N+1)/(N+2)})$ regret is achievable.

## 1.3. Related Work

The principal-agent model has been extensively studied in both one-shot and repeated settings. In the one-shot setting (Holmström, 1979; Grossman & Hart, 1992; Carroll, 2015; Dütting et al., 2019; 2022), the principal only interacts with agents once, and the typical objective is to identify the best

---

[1] We use $\tilde{\Omega}(\cdot), \widetilde{\mathcal{O}}(\cdot)$ to suppress poly-logarithmic terms in $K, T$.

(approximately) incentive. In the *repeated* setting (Misra et al., 2005; Misra & Nair, 2011; Ho et al., 2014; Kaynar & Siddiq, 2023; Scheid et al., 2024b; Dogan et al., 2023a; Ben-Porat et al., 2024; Wu et al., 2024; Liu & Ratliff, 2024), the principal interacts with agents for multiple rounds, and the goal could be estimating agents' model or maximizing the cumulative profits, which in turn minimize the regret. Our work falls within this line of research. While principal-multi-agent problems have been considered in the one-shot setup (Dütting et al., 2023; Duetting et al., 2025), little is known about the multi-agent study in the repeated setting. The known related works (Ho et al., 2014; Gayle & Miller, 2015) consider the repeated principal-multi-agent problems, but they assume that the arrival of agents follow a fixed underlying distribution, which is often not the case in many real-world scenarios.

The repeated principal-agent problem has been studied under different forms of information asymmetry. The moral hazard setting (Misra et al., 2005; Ho et al., 2014; Kaynar & Siddiq, 2023) assumes that the agents' actions are invisible to the principal, while the adverse selection (Ho et al., 2014; Gayle & Miller, 2015) considers cases where agents' preferences are unknown to the principal. Additionally, Scheid et al. (2024b;a); Dogan et al. (2023a); Liu & Ratliff (2024) explore settings where the agent's type is unknown, but they assume a single agent type in the interaction, avoiding the challenge of multiple agent types responding differently to the same incentive—a key issue in our work. Fiez et al. (2018) study agents with multiple different types that evolve dynamically (as a function of the offered incentives) according to a controlled Markov chain, and focus their attention on epoch-based algorithms that exploit mixing time properties of the controlled Markov chain along with a greedy matching strategy in each epoch.

While our framework shares the principal–agent structure of Harris et al. (2024) and Balcan et al. (2025), it diverges from their contextual Stackelberg models in two key respects. First, in their setting, the principal chooses a mixed strategy from a probability simplex over a finite action set, whereas in our setting, the principal chooses an incentive vector from the hypercube $\pi \in [0,1]^N$. In addition, their payoffs of the principal and agent derive from a contextual utility function

evaluated under the chosen mixed strategy and observed context. In contrast, in our model each agent of unknown type $j$ selects arm $i \in [N]$ to maximize $\mu_i^j + \pi_i$, and the principal's payoff is $v_i - \pi_i$. These key differences in action space and reward structure highlight the distinction between our principal-agent model and theirs.

Moreover, similar to the works of Bernasconi et al. (2023) and Balcan et al. (2025), our algorithm proceeds by cleverly reducing the problem to an online linear optimization problem.

# 2. Lower Bounds for Greedy Choice Model

In this section, we establish the fundamental limits of the principal-agent problem under the single-arm incentive. We first show in Section 2.1 that any algorithm lacking advance knowledge of the agents' behavior can be forced to incur linear regret. Then, in Section 2.2, when the agents' behaviors are known, we derive a tight lower bound on regret for the greedy choice model, up to logarithmic factors.

## 2.1. Linear Regret for Unknown Agent Behavior

Consider the greedy choice model, but now assume that the principal does not have access to the best response function $b(\cdot, \cdot)$. Under this assumption, we establish a linear regret bound for any algorithm, as stated below.

**Theorem 2.1.** *Suppose the principal and the agents operate under a greedy choice model with single-arm incentives, and the principal does not have access to the best response function $b(\cdot, \cdot)$. For any algorithm, there exists an instance of the principal-agent problem with $K = 2$ agent types and $N = 3$ arms such that $R_T = \Omega(T)$.*

To provide intuition for Theorem 2.1, consider the following example:

**Example 2.2.** *The principal's reward vector is $v = (1, 0.5, 0)$. The preference vectors of two agents are $\mu^1 = (0.2, 0, 0.2 + \Delta)$ and $\mu^2 = (0.2, 0.2 + \Delta, 0)$, where $\Delta \in [0.7, 0.71]$. When a tie occurs, agent 1 favors arm 1, while agent 2 favors arm 2. At each round, agent 1 is selected with probability $0.4$, and agent 2 is selected with probability $0.6$.*

In Example 2.2, the incentive vector $\pi^* = (\Delta, 0, 0)$ is uniquely optimal because any deviation from this incentive for arm 1 incurs at least a constant regret. Specifically, the expected reward of $\pi^*$ is $0.6 \cdot 0.5 + 0.4(1 - \Delta) \geq 0.416$. In contrast, offering an incentive less than $\Delta$ for arm 1 results in an expected reward of at most $0.6 \cdot 0.5 + 0.4 \cdot 0 = 0.3$, and offering an incentive greater than $\Delta$ yields an expected reward of at most $1 - \Delta \leq 0.3$.

However, the probabilistic selection of agents implies that we must exactly learn $\Delta$ to achieve sub-linear regret, which

is not always feasible. Consequently, any algorithm incurs an $\Omega(T)$ regret for some $\Delta \in [0.7, 0.71]$. For the formal proof, we refer the reader to Appendix B.

## 2.2. Lower Bounds for Single-arm Incentive

Consider the greedy choice model, where the principal is restricted to single-arm incentives and has knowledge of the best response function $b(\cdot, \cdot)$. The following theorem establishes a tight lower bound on regret, up to logarithmic factors.

**Theorem 2.3.** *Suppose the principal and the agents operate under a greedy choice model with single-arm incentives, and the principal has access to the best response function $b(\cdot, \cdot)$. For any $K \geq 3$, $N \geq 3$, $T \geq poly(K)$, and any algorithm, there exists an instance of the principal-agent problem such that*

$$R_T = \Omega \left( \min \left\{ \sqrt{KT \log(N)/\log(K)}, K\sqrt{T/\log(K)} \right\} \right)$$

The proof of this theorem is non-trivial, and due to space constraints, we defer it to Appendix C. Here, we outline the key ideas behind our proof.

First, consider the case where $N \leq K$. We carefully construct the reward vector $v$ and the preference vectors $\mu^j$ such that there exists a set $\Pi = \{\pi^{(1)}, \pi^{(2)}, \ldots, \pi^{(K-1)}\}$ of incentive vectors with the property that for any $\pi \in \mathcal{D}$, there exists some $\pi^{(i)} \in \Pi$ satisfying $U(\pi^{(i)}, j) \geq U(\pi, j)$ for all $j \in [K]$. Consequently, we can assume that any algorithm selects an incentive exclusively from $\Pi$ in each round.

Next, we define a probability distribution $p^{(1,\varepsilon)}$ over the agents, ensuring that in each round, an agent $j_t$ is drawn from this fixed distribution. We construct $p^{(1,\varepsilon)}$ such that the expected utility of choosing the incentive vector $\pi^{(1)}$ is $r + \varepsilon$, while for all other incentive vectors, it remains $r$, where $r \in [0, 1/2]$ is some constant.

Now, suppose the algorithm selects an incentive vector $\pi^{(i)} \neq \pi^{(1)}$ only a limited number of times. We construct an alternate problem instance by instead using a probability distribution $p^{(i,\varepsilon)}$, where the agent $j_t$ is sampled from $p^{(i,\varepsilon)}$ in each round. This new distribution is chosen so that the expected utility of $\pi^{(1)}$ remains $r + \varepsilon$, while that of $\pi^{(i)}$ increases to $r + 2\varepsilon$, and all other incentive vectors continue to yield $r$.

By appropriately choosing $\varepsilon$ and employing careful KL-divergence-based arguments, we establish that in at least one of these two problem instances, the algorithm incurs a regret of $\Omega(\sqrt{KT})$. While these arguments resemble the standard lower bound analysis for the Multi-Armed Bandit (MAB) problem, a key challenge in our setting is that we must fix the reward vector and preference vectors in

advance, constructing multiple instances solely by varying the probability distributions over the agents.

For the case when $N \geq K$, we aim to establish a combinatorial bandit-style lower bound, analogous to the MAB-style lower bound derived for $N \leq K$. At a high level, we begin by defining a bijective mapping $f$ that maps the first $N-1$ arms to a subset of $\{0,1\}^{K-2}$. These $N-1$ arms each yield a reward of roughly 0.5, while the last arm is a special arm with a reward of 0.

We then construct preference vectors for each agent such that there exists a set $\Pi = \{\pi^{(1)}, \pi^{(2)}, \ldots, \pi^{(N-1)}\}$ of incentive vectors satisfying the following property: for any $\pi \in \mathcal{D}$, there exists some $\pi^{(i)} \in \Pi$ such that $U(\pi^{(i)}, j) \geq U(\pi, j)$ for all $j \in [K]$. Also for all $i \in [N-1]$, $(\pi^{(i)})_i > 0$. Consequently, we can assume that any algorithm selects an incentive exclusively from $\Pi$ in each round.

Next, we define a class of distributions for sampling agents such that the utility function exhibits a linear structure over the vectors in $\Pi$. Specifically, we ensure that

$$\mathbb{E}_{j \sim p}[U(\pi^{(i)}, j)] = \langle f(i), \theta^{(p)} \rangle,$$

where $\theta^{(p)} \in [0,1]^{K-2}$ is a reward vector dependent solely on the preference vectors of the agents and the distribution $p$ from which the agents are sampled in each round. The last arm and two special agents who prefer this arm play a crucial role in ensuring the above equality.

Given this setup, we replicate a combinatorial bandit-style lower bound by carefully defining the sample space and computing the corresponding KL-divergences. Ultimately, we establish a lower bound of

$$\Omega\left(\min\left\{\sqrt{KT \log(N)/\log(K)}, K\sqrt{T/\log(K)}\right\}\right).$$

## 3. Main Results for Greedy Choice Model

In this section, we present our algorithms for the greedy choice model, assuming access to the agents' best response functions. The key idea behind our algorithm design is to discretize the decision space and reduce the problem to adversarial linear bandits. We first describe our algorithm for the single-arm incentive in Section 3.1, followed by the algorithm for the general incentive in Section 3.2.

### 3.1. Algorithm for Single-arm Incentive

The proposed algorithm consists of two stages: first, discretizing the decision space $\mathcal{D} = \{x \in [0,1]^N : |\text{supp}(x)| \leq 1\}$ into a finite set, and second, reducing the problem to adversarial linear bandits using the discretized set and the best response functions of each agent.

**Tie Breaking.** For simplicity of presentation, we assume that in the case of a tie, an agent always prefers the incentivized arm, and each preference vector has a unique

maximum. Our approach can be suitably modified to avoid relying on this assumption. For further details, we refer the reader to Appendix D.

**Discretization.** To construct a finite single-arm incentive set, we begin with an empty set $\Pi$. Observe that $\max_{k \in [N]} \mu_k^j - \mu_i^j$ represents the minimum incentive required to entice agent $j$ to choose arm $i$. For each arm $i \in [N]$ and agent $j \in [K]$, we define an incentive vector $\pi^{i,j} \in \mathcal{D}$ such that for all $s \in [N]$:

$$(\pi^{i,j})_s = \begin{cases} \max_{k \in [N]} \mu_k^j - \mu_i^j, & \text{if } s = i, \\ 0, & \text{otherwise.} \end{cases}$$

For all $i \in [N]$ and $j \in [K]$, we add the incentive vector $\pi^{i,j}$ to $\Pi$, resulting in $|\Pi| = \mathcal{O}(NK)$. The set $\Pi$ has the key property that for any $\pi \in \mathcal{D}$, there exists a vector $\pi^{i,j} \in \Pi$ such that $U(\pi^{i,j}, \widehat{j}) \geq U(\pi, \widehat{j})$ for all $\widehat{j} \in [K]$.

When $N = \Omega(2^K)$ is exponentially large, we can further refine the discretized set to make it independent of $N$. Intuitively, when $N$ is large, some incentive vectors elicit the same response in terms of the selection of their incentivized arms. In such cases, we retain only the most rewarding incentive vector rather than including all equivalent ones in the reduced set.

To formalize this, we define a mapping $h : \Pi \to \{0,1\}^K$ such that for any $\pi \in \Pi \setminus \{\mathbf{0}\}$ and $j \in [K]$, where $\mathbf{0}$ is the zero-incentive vector $(0, 0, \ldots, 0)$, we have:

$$(h(\pi))_j = \begin{cases} 1, & \text{if } b(\pi, j) = a(\pi), \\ 0, & \text{otherwise,} \end{cases}$$

where $a(\pi) := \arg\max_{i \in [N]} \pi_i$ denotes the index of the largest coordinate of $\pi$. Observe that if $b(\pi, j) \neq a(\pi)$ then $b(\pi, j) = b(\mathbf{0}, j)$.

Using this mapping, we construct a reduced set $\widehat{\Pi}$ as follows. We initialize $\widehat{\Pi}$ as an empty set. For each vector $s \in \{0,1\}^K$, define

$$\Pi_s := \{\pi \in \Pi \setminus \{\mathbf{0}\} : h(\pi) = s\}.$$

If $|\Pi_s| = 1$, we add the unique $\pi \in \Pi_s$ to $\widehat{\Pi}$. If $|\Pi_s| > 1$, we have $U(\pi, j) = s_j \cdot (v_{a(\pi)} - \pi_{a(\pi)}) + (1 - s_j) \cdot v_{b(\mathbf{0}, j)}$ for all $\pi \in \Pi_s$. Thus, we add only one of the maximizers, $\hat{\pi} = \arg\max_{\pi \in \Pi_s} v_{a(\pi)} - \pi_{a(\pi)}$, to $\widehat{\Pi}$. We also include the zero-incentive vector $\mathbf{0}$ in $\widehat{\Pi}$. Since $|\{0,1\}^K| = 2^K$ and we select at most one vector per $\Pi_s$, we obtain $|\widehat{\Pi}| \leq \min\{2KN, 2^K\} + 1$.

Since $\widehat{\Pi}$ is obtained by removing only suboptimal incentive vectors from $\Pi$, it retains the key property that for any $\pi \in \mathcal{D}$, there exists $\pi^{i,j} \in \widehat{\Pi}$ satisfying $U(\pi^{i,j}, \widehat{j}) \geq U(\pi, j)$ for all $\widehat{j} \in [K]$. Given this fact, it suffices to focus only on $\widehat{\Pi}$ for regret minimization.

A natural approach is to treat each incentive $\pi \in \widehat{\Pi}$ as an arm, thereby reducing the problem to an adversarial multi-armed bandit setting. However, this reduction results in a regret bound of $\mathcal{O}(\sqrt{\min\{KN, 2^K\}T})$ if one, for example, applies the Tsallis-INF algorithm (Zimmert & Seldin, 2021).

Given the lower bound established in Section 2.2, this raises the question: can we adopt a different approach to improve the upper bound?

**Reduction to Adversarial Linear Bandits.** We now improve the upper bound by proposing a simple yet effective reduction to the adversarial linear bandits problem. This approach ensures that our upper bound matches the lower bound up to logarithmic factors in $K$.

We begin by constructing an arm set $\mathcal{Z} \subseteq \mathbb{R}^K$. For each incentive $\pi \in \widehat{\Pi}$, we define the corresponding vector $z^\pi \in \mathbb{R}^K$ as

$$(z^\pi)_j = U(\pi, j), \quad \forall j \in [K].$$

We then add $z^\pi$ to $\mathcal{Z}$. Next, we define the reward vector $y_t \in \mathbb{R}^K$. If agent $j_t$ arrives in round $t$, we set

$$(y_t)_j = \begin{cases} 1, & \text{if } j = j_t, \\ 0, & \text{otherwise.} \end{cases}$$

With these definitions, for any $\pi \in \widehat{\Pi}$, we observe that $U(\pi, j_t) = \langle z^\pi, y_t \rangle$. Thus, our pseudo-regret $R_T$ is equal to the adversarial linear bandit pseudo-regret:

$$R_T = \max_{z \in \mathcal{Z}} \mathbb{E} \left[ \sum_{t=1}^T \langle z, y_t \rangle - \sum_{t=1}^T \langle z_t, y_t \rangle \right].$$

For an adversarial linear bandit problem over a decision set $\mathcal{X} \subset \mathbb{R}^d$ with horizon $T$ and rewards bounded in $[-1, 1]$, let $R_{\mathbf{Alg}}(T, \mathcal{X})$ denote the pseudo-regret of an algorithm **Alg**. If **Alg** is the EXP3 algorithm for linear bandits (Lattimore & Szepesvári, 2020), it satisfies

$$R_{\mathbf{Alg}}(T, \mathcal{X}) \leq \mathcal{O}(\sqrt{dT \log(|\mathcal{X}|)}).$$

Since our reduction is an instance of the adversarial linear bandit problem, we obtain the following result:

**Theorem 3.1.** *Suppose the principal and the agents operate under a greedy choice model with single-arm incentives, and the principal has access to the best response function $b(\cdot, \cdot)$. Then, there exists an algorithm for this principal-agent problem whose pseudo-regret is upper bounded by*

$$R_T \leq \tilde{\mathcal{O}} \left( \min \left\{ \sqrt{KT \log(N)}, K\sqrt{T} \right\} \right).$$

### 3.2. Algorithm for General Incentive

The proposed algorithm consists of two stages: first, discretizing the decision space $\mathcal{D} = [0, 1]^N$ into a finite set,

and second, reducing the problem to adversarial linear bandits using the discretized set and the best response functions of each agent.

**Tie Breaking.** To simplify the presentation, let us assume that agents resolve ties in a hierarchical manner. Specifically, each agent $j$ has a bijective mapping $f_j : [N] \to [N]$, and in the event of a tie among a set of arms $\mathcal{I} \subseteq [N]$, agent $j$ selects the arm $\arg\max_{i \in \mathcal{I}} f_j(i)$. Other tie-breaking rules can be handled in a similar fashion.

**Discretization.** For the general incentive case, the principal's decision space is given by $\mathcal{D} = [0, 1]^N$. The algorithm for this case builds upon the reduction framework employed by the single-arm incentive algorithm but introduces a different discretization approach.

Let $\sigma \in [N]^K$ be a $K$-dimensional tuple. Define $\mathcal{P}_\sigma := \{\pi \in [0, 1]^N : b(\pi, j) = \sigma_j \ \forall j \in [K]\}$, and let $\widehat{\mathcal{P}}_\sigma$ denote the closure of $\mathcal{P}_\sigma$. If $\widehat{\mathcal{P}}_\sigma$ is non-empty, then it forms a convex polytope defined by the following set of linear inequalities:

$$0 \leq \pi_i \leq 1 \quad \forall i \in [N],$$

$$\mu_{\sigma_j}^j + \pi_{\sigma_j} \geq \mu_i^j + \pi_i \quad \forall i \in [N] \setminus \{\sigma_j\}, \ j \in [K].$$

If $\mathcal{P}_\sigma$ is non-empty, it is an open convex polytope, as some of the inequalities above become strict.

Now, let $\Sigma := \{\sigma \in [N]^K : \mathcal{P}_\sigma \neq \emptyset\}$. We define two incentive vectors $\pi$ and $\hat{\pi}$ to be $\varepsilon$-close if $\|\pi - \hat{\pi}\|_\infty \leq \varepsilon$. For $\sigma \in \Sigma$, we construct a set of incentive vectors $\Pi_\sigma$ as follows: for each extreme point of $\widehat{\mathcal{P}}_\sigma$, we select an $\varepsilon$-close vector in $\mathcal{P}_\sigma$ and include it in $\Pi_\sigma$. We now claim that $|\Pi_\sigma| \leq \text{poly}(N, K)^N$.

First note that any extreme point of $\widehat{\mathcal{P}}_\sigma$ is an intersection of $N$-linearly independent half-spaces defining this polytope. Now observe that the number of half spaces used to define this polytope is at most $\text{poly}(N, K)$. Hence, the number of extreme points is upper bounded by $(\text{poly}(N, K))^N$. Since each point in $\Pi_\sigma$ is associated with exactly one extreme point of $\widehat{\mathcal{P}}_\sigma$, we have $|\Pi_\sigma| \leq (\text{poly}(N, K))^N$.

Let $\Pi := \bigcup_{\sigma \in \Sigma} \Pi_\sigma$. We claim that for any sequence of agents $j_1, j_2, \ldots, j_T$, the following inequality holds:

$$\max_{\pi \in \Pi} \sum_{t=1}^T U(\pi, j_t) \geq \sup_{\pi \in [0,1]^N} \sum_{t=1}^T U(\pi, j_t) - 2\varepsilon T.$$

Consider an incentive vector $\pi^\star \in [0, 1]^N$ such that

$$\sum_{t=1}^T U(\pi^\star, j_t) \geq \sup_{\pi \in [0,1]^N} \sum_{t=1}^T U(\pi, j_t) - \varepsilon T.$$

Let $\pi^\star \in \mathcal{P}_\sigma$, where $\sigma$ corresponds to the behavior induced by $\pi^\star$. Since the maximum of a linear function over a closed

polytope is attained at one of its extreme points, there exists an extreme point $\widehat{\pi}$ of $\widehat{\mathcal{P}}_\sigma$ such that

$$\sum_{t=1}^{T} v_{\sigma_{j_t}} - \widehat{\pi}_{\sigma_{j_t}} \geq \sum_{t=1}^{T} v_{\sigma_{j_t}} - \pi^\star_{\sigma_{j_t}} = \sum_{t=1}^{T} U(\pi^\star, j_t).$$

Let $\tilde{\pi}$ be the vector in $\Pi_\sigma$ that is $\varepsilon$-close to $\widehat{\pi}$. Now, we have the following:

$$\begin{aligned}
\sum_{t=1}^{T} U(\tilde{\pi}, j_t) &= \sum_{t=1}^{T} v_{\sigma_{j_t}} - \tilde{\pi}_{\sigma_{j_t}} \\
&= \sum_{t=1}^{T} (v_{\sigma_{j_t}} - \widehat{\pi}_{\sigma_{j_t}}) - \sum_{t=1}^{T} (\tilde{\pi}_{\sigma_{j_t}} - \widehat{\pi}_{\sigma_{j_t}}) \\
&\geq \sum_{t=1}^{T} (v_{\sigma_{j_t}} - \widehat{\pi}_{\sigma_{j_t}}) - \sum_{t=1}^{T} |\tilde{\pi}_{\sigma_{j_t}} - \widehat{\pi}_{\sigma_{j_t}}| \\
&\geq \sum_{t=1}^{T} (v_{\sigma_{j_t}} - \widehat{\pi}_{\sigma_{j_t}}) - \varepsilon T \\
&\qquad\qquad (\text{as } ||\widehat{\pi} - \tilde{\pi}||_\infty \leq \varepsilon) \\
&\geq \sup_{\pi \in [0,1]^N} \sum_{t=1}^{T} U(\pi, j_t) - 2\varepsilon T
\end{aligned}$$

If we set $\varepsilon = \frac{1}{T}$, then it suffices to focus on the set of incentive vectors $\Pi$ for our regret minimization problem.

**Reduction to Adversarial Linear Bandits.** We now show a reduction of this problem to an adversarial linear bandit problem. First, we construct a set $\mathcal{Z} \subset \mathbb{R}^K$ as follows: for each $\pi \in \Pi$, we add $z^\pi$ to $\mathcal{Z}$, where $(z^\pi)_j = U(\pi, j)$ for all $j \in [K]$. Next, we define the reward vector $y_t \in \mathbb{R}^K$. If agent $j_t$ arrives in round $t$, we set $(y_t)_j = 1$ if $j = j_t$ and $(y_t)_j = 0$ otherwise. Observe that for any $\pi \in \Pi$, $U(\pi, j_t) = \langle z^\pi, y_t \rangle$. Hence, our pseudo-regret $R_T$ is equal to the adversarial linear bandit pseudo-regret: :

$$R_T = \max_{z \in \mathcal{Z}} \mathbb{E}\left[\sum_{t=1}^{T} \langle z, y_t \rangle - \sum_{t=1}^{T} \langle z_t, y_t \rangle\right].$$

Using the EXP3 algorithm for linear bandits, we obtain a regret upper bound of $\text{poly}(K, N) \cdot \sqrt{T}$ for our regret minimization problem. However, this upper bound can be further improved as follows. Let $\mathcal{R} = \{e_j\}_{j \in [K]}$ be the set of all possible reward vectors. Define $\mathcal{Z}_0$ as the smallest subset of $\mathcal{Z}$ such that

$$\max_{z \in \mathcal{Z}} \min_{z' \in \mathcal{Z}_0} \max_{y \in \mathcal{R}} |\langle z - z', y \rangle| \leq \frac{1}{T}.$$

By the above inequality, it suffices to focus on $\mathcal{Z}_0$ for our regret minimization problem. It can be shown that $|\mathcal{Z}_0| \leq (6KT)^K$ (see Chapter 27 of Lattimore & Szepesvári

(2020)). Using the EXP3 algorithm for linear bandits on the reduced arm set $\mathcal{Z}_0$, we achieve a regret upper bound of $\mathcal{O}\left(K\sqrt{T \log(KT)}\right)$. This leads to the following theorem:

**Theorem 3.2.** *Suppose the principal and agents operate under a greedy choice model with general incentives, and the principal has access to the best response function $b(\cdot, \cdot)$. Then, there exists an algorithm for this principal-agent problem whose pseudo-regret is upper bounded by*

$$R_T \leq \mathcal{O}\left(K\sqrt{T \log(KT)}\right).$$

# 4. Main Results for Smooth Choice Model

In this section, we present an algorithm for the setting where agent types and their preference model are unknown but assumed to be a smooth function of the incentive vector.

## 4.1. Algorithm for Single-Arm Incentives

The proposed algorithm comprises two stages. First, it discretizes the decision space $\mathcal{D} = \{x \in [0,1]^N : |\text{supp}(x)| \leq 1\}$ into a finite set. Second, it runs the adversarial multi-armed bandit algorithm, Tsallis-INF, on the discretized set.

**Discretization.** Fix $\varepsilon > 0$. To construct a finite single-arm incentive set, we begin with an empty set $\Pi$. For all $i \in [N]$ and $j \in [1/\varepsilon + 1]$, we define an incentive vector $\pi^{i,j} \in \mathcal{D}$ such that for all $s \in [N]$:

$$(\pi^{i,j})_s = \begin{cases} (j-1) \cdot \varepsilon, & \text{if } s = i, \\ 0, & \text{otherwise.} \end{cases}$$

**Algorithm and its Regret guarantee.** We run Tsallis-INF by treating each vector in $\Pi$ as an arm. Let $\pi^* \in \mathcal{D}$ be the optimal fixed incentive in hindsight against a sequence of agents $j_1, j_2, \ldots, j_T$. Consider a vector $\widehat{\pi} \in \Pi$ such that $|\widehat{\pi}_{\hat{i}} - \pi^*_{\hat{i}}| \leq \varepsilon$ for some index $\hat{i} \in [N]$. For any agent $j$, we have the following:

$$\begin{aligned}
&\mathbb{E}[U(\pi^*, j)] - \mathbb{E}[U(\widehat{\pi}, j)] \\
&= \sum_{i=1}^{N} (Pr[a(\pi^*, j) = i] - Pr[a(\widehat{\pi}, j) = i]) \cdot v_i \\
&\quad + Pr[a(\widehat{\pi}, j) = i] \cdot \widehat{\pi}_{\hat{i}} - Pr[a(\pi^*, j) = i] \cdot \pi^*_{\hat{i}} \\
&\leq \sum_{i=1}^{N} (Pr[a(\pi^*, j) = i] - Pr[a(\widehat{\pi}, j) = i]) \cdot v_i \\
&\quad + (Pr[a(\widehat{\pi}, j) = i] - Pr[a(\pi^*, j) = i]) \cdot \pi^*_{\hat{i}} + \varepsilon \\
&\leq 2 \cdot \varepsilon \cdot L + \varepsilon \quad (\text{Assumption 1.1 and } 0 \leq v_i, \pi^*_{\hat{i}} \leq 1)
\end{aligned}$$

The regret of our approach is upper bounded as:

$$R_T \leq \sum_{i=1}^{T} \mathbb{E}[U(\widehat{\pi}, j_t) - U(\pi_t, j_t)]$$

$$+ \sum_{i=1}^{T} \mathbb{E}[U(\pi^*, j_t) - U(\widehat{\pi}, j_t)]$$

$$\leq \mathcal{O}(\sqrt{N\epsilon^{-1}T} + T \cdot (2L+1) \cdot \epsilon).$$

Setting $\varepsilon = N^{1/3}(2L+1)^{-2/3}T^{-1/3}$, would make the regret bounded by $\mathcal{O}((2L+1)^{1/3}N^{1/3}T^{2/3})$.

**Remark:** It can be easily shown that without knowing $L$, any fixed discretization's regret will scales linearly in $L$.

## 4.2. Lower Bound for Single-Arm Incentives

In the following theorem, we show the our regret upper bound is optimal.

**Theorem 4.1.** *For all $N, T$, and $L \geq 3$, if Assumption 1.1 holds for all agents, then we have*

$$\inf_{\mathcal{A} \in Alg} \sup_{j \in J} \mathbb{E}[R_T^{\mathcal{A},j}] = \Omega\left(L^{1/3}N^{1/3}T^{2/3}\right),$$

*where $J$ is the set of agent types, and $R_T^{\mathcal{A},j}$ is the regret when only agents of type $j$ are selected.*

Here we provide high level ideas behind the proof. We refer the reader to Appendix E for the detailed proof. To prove the above theorem, we construct an instance where only a small interval of size $\epsilon$ for a specific arm yields a reward $\Delta$ higher than any other incentive-arm pair. This results in a regret of $\min(\Delta T, \Delta^{-2}\epsilon^{-1}N)$, which, with appropriately chosen parameters, establishes the desired bound.

However, proving this bound presents two additional technical challenges, as Assumption 1.1 must hold. Specifically, the reward within the optimal interval should increase smoothly by $\Delta$ and then return to the base reward. Additionally, $\Delta$ must be bounded by $\mathcal{O}(L\epsilon^{-1})$. We define the set of adversarial types as $J := \{(i,j) \mid i \in [N], j \in \frac{1}{2}\lfloor\epsilon^{-1}\rfloor\}$. Then, for all $l \leq N-1$, we have:

$$\Pr[a(\pi, (i,j)) = l] = \begin{cases} \frac{1}{16N(1-\pi_i)} + \mathbf{B}(\pi_i - j\epsilon) \\ \quad \text{if } l = i, \pi_i \in [j\epsilon, (j+1)\epsilon] \\ \frac{1}{16N(1-\pi_i)} & \text{if } \pi_l \leq \frac{1}{2} \\ \frac{1}{8N} & \text{if } \pi_l > \frac{1}{2} \end{cases} \tag{1}$$

$$\Pr[a(\pi, (i,j)) = N] = 1 - \sum_{l=1}^{N-1} \Pr[a(\pi, (i,j)) = l] \tag{2}$$

Where bonus term $\mathbf{B} : [0, \epsilon] \to \mathbb{R}$ is a $\frac{L-1}{4}$-Lipschitz function with $\mathbf{B}(0) = \mathbf{B}(\epsilon) = 0$. This leads to the optimal incentive $\pi^*$, where $\pi_l^* = \begin{cases} j\epsilon + \arg\max_x \mathbf{B}(x) & \text{if } l = i \\ 0 & \text{if } l \neq i \end{cases}$, and only giving incentive in a small region containing $\pi^*$ will give information for optimal solution, and its not possible to distinguish the best region for all agent types without having $\Omega(T^{2/3})$ regret.

## 4.3. Instance-dependent Algorithm for Single-Arm Incentives

Depending on the problem instance, uniformly covering the entire action space—while optimal in the worst case—can be inefficient. The goal is to cover near-optimal regions more densely than others. To achieve this, we leverage zooming algorithms, which are widely used in the Lipschitz bandits literature. These algorithms begin with a uniform discretization and adaptively refine the discretization in regions that are more likely to be optimal. Specifically, if the uncertainty about an optimal point being in a given region falls below a certain threshold (the zooming rule), that region is discretized more densely.

We use ADVERSARIALZOOMING from (Podimata & Slivkins, 2021) with the action set $\mathcal{D}$ and the metric $d(\pi, \pi') = L\|\pi - \pi'\|_{\infty}$. If Assumption 1.1 holds, then the reward function $U(\cdot, j_t)$ is $(2L+1)$-Lipschitz, ensuring that these inputs are valid for the algorithm. To adapt to the adversarial setting, the algorithm employs EXP3.P to sample points from active regions and zooms into a region when its confidence interval becomes smaller than a threshold, determined by its diameter, $L$, and $t$. The key instant-dependent parameter is the adversarial zooming dimension, for a parameter $\gamma > 0$ is defined as:

$$\inf_{z \geq 0} \{\text{Cover}(\mathcal{A}_{\varepsilon}, \mathcal{B}(\|.\|_{\infty}, \varepsilon)) \leq \gamma\varepsilon^{-z} \quad \forall\varepsilon\},$$

where $\mathcal{A}_{\varepsilon}$ is the set of all incentives that achieve $\widetilde{\mathcal{O}}(\varepsilon)$-optimal reward for the principal in hindsight, and $\mathcal{B}(\|\|, \varepsilon)$ are (noncentered) norm balls of diameter $\varepsilon$. Then, using (Podimata & Slivkins, 2021)[Theorem 3.1] for our setting, we would have the following regret bound.

**Corollary 4.2.** *Suppose the principal and agents operate under a smooth choice model (Assumption 1.1), where the choice model has the Adversarial zooming dimension of $z$. Then, there exists an algorithm for this principal-agent problem whose pseudo-regret is upper bounded by*

$$R_T \leq \mathcal{O}(T^{(z+1)/(z+2)})((2L+1)N)^{z/(z+2)} \log^5 T.$$

For single-incentive setting, the zooming dimension is bounded by 1, since the entire action space can be covered with $\mathcal{O}(N\epsilon^{-1})$ hypercubes, as shown in the fixed discretization algorithm. However, for some smooth choice models, the regret bound is improved. For instance, if the reward function induced by an agent's choice model is strictly concave, then $z = \frac{1}{2}$, leading to an expected regret of $T^{3/5}$ (see (Podimata & Slivkins, 2021), Section 6).

## 4.4. Algorithm with General Incentives

For the general incentive case, recall that $\mathcal{D} = [0,1]^N$. We partition $\mathcal{D}$ into $\varepsilon^{-N}$ equally sized hypercubes and define

$\Pi$ as the set of center points of these hypercubes. Similar to the single-arm incentive case, we run Tsallis-INF by treating each point in $\Pi$ as an arm. Setting $\varepsilon = (2L + 1)^{-2/(N+2)}T^{-1/(N+2)}$, we incur the following regret:

$$
\begin{aligned}
R_T &\leq \sum_{i=1}^{T} \mathbb{E}[U(\widehat{\pi}, j_t) - U(\pi_t, j_t)] \\
&\quad + \sum_{i=1}^{T} \mathbb{E}[U(\pi^*, j_t) - U(\widehat{\pi}, j_t)] \\
&\leq \mathcal{O}(\sqrt{\varepsilon^{-N}T} + T \cdot (2L+1) \cdot \varepsilon) \\
&\leq \mathcal{O}\left((2L+1)^{N/(N+2)} T^{(N+1)/(N+2)}\right)
\end{aligned}
$$

Similar to the single-incentive setting, using ADVERSARIALZOOMING will result in a regret of $\widetilde{\mathcal{O}}((2L+1)^{z/(z+2)}T^{(z+1)/(z+2)})$, where the zooming dimension $z$ is upper bounded by $N$, as $\epsilon^{-N}$ hypercubes cover all of decision space.

## 5. Conclusion

In this paper, we initiate the study of the repeated principal-agent problem with adversarial agent arrivals. For the greedy response agent and single-arm incentive case, we establish a negative result, proving that any algorithm without prior knowledge of agents' behaviors can be forced to incur linear regret. However, when the agents' behaviors are known, we design an algorithm based on a novel reduction to adversarial linear bandits, achieving a regret bound that nearly matches the lower bound up to a logarithmic factor. Moreover, when the principal is allowed to offer general incentives, we propose an algorithm that integrates our reduction technique with a novel discretization approach, leading to a sublinear regret bound. Finally, for the smooth-response agent setting, we develop an algorithm whose regret bound aligns with our lower bound up to logarithmic factors.

Our work opens several interesting directions for future research. A natural extension, inspired by (Scheid et al., 2024b), is to consider a setting where the principal observes only noisy rewards, adding an extra layer of uncertainty and requiring reward distribution learning. Another direction is to incorporate purchase quantity into the model, as in (Chen et al., 2024). While our current framework assumes agents purchase a single item per round, real-world scenarios often involve varying purchase quantities.

## ACKNOWLEDGEMENTS

This work was supported in part by NSF TRIPODS CCF Award #2023166, Microsoft Grant for Customer Experience Innovation, a Northrop Grumman University Research Award, ONR YIP award # N00014-20-1-2571, and NSF award #1844729.

## Impact Statement

This paper presents work whose goal is to advance the field of Machine Learning. There are many potential societal consequences of our work, none which we feel must be specifically highlighted here.

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

## A. Technical Lemmas

**Lemma A.1** (KL-divergence of Bernoulli ([Maiti et al. (2023)](#))). *Consider* $\varepsilon > 0$, $\varepsilon < c_0 \leq \frac{1}{2}$ *such that* $\frac{\varepsilon}{c_0} \leq \frac{1}{2}$. *Let* $P$ *and* $Q$ *be bernoulli distributions with means* $c_0 - \varepsilon$ *and* $c_0 + \varepsilon$. *Then we have the following:*

$$\mathrm{KL}(P, Q) \leq \frac{16\varepsilon^2}{c_0} \quad and \quad \mathrm{KL}(Q, P) \leq \frac{16\varepsilon^2}{c_0}$$

**Lemma A.2** (Chain Rule ([Maiti et al. (2025)](#))). *Let* $f(x_1, x_2, \ldots, x_n)$ *and* $g(x_1, x_2, \ldots, x_n)$ *be two joint PMFs for a tuple of random variables* $(X_i)_{i \in [n]}$. *Let the sample space be* $\Omega = \{0, 1\}^n$. *Then we have the following:*

$$\mathrm{KL}(f, g) = \sum_{\omega \in \Omega} f(\omega) \left( \mathrm{KL}(f(X_1), g(X_1)) + \sum_{i=2}^{n} \mathrm{KL}(f(X_i | X_{-i} = \omega_{-i}), g(X_i | X_{-i} = \omega_{-i})) \right)$$

*where* $X_{-i} = (X_1, \ldots, X_{i-1})$, $\omega_{-i} = (\omega_1, \ldots, \omega_{i-1})$.

## B. $\Omega(T)$ Lower Bound for unknown behavior of the agents

Let us consider the setting where the principal doesn't know the behavior of the agents. In this case, we show that the principal necessarily incurs a regret of $\Omega(T)$ under the single arm incentive. Let us choose $\Delta \in [0.7, 0.71]$ uniformly at random. Let there be two agent types. Let $u^j$ be the reward vector associated with each agent $j \in \{1, 2\}$. We define $u_1^1 = 0.2, u_2^1 = 0$ and $u_3^1 = 0.2 + \Delta$. We also define $u_1^2 = 0.2, u_2^2 = 0.2 + \Delta$ and $u_3^2 = 0$. For an incentive of $\Delta$ to arm 1, agent 1 breaks the tie in favor of arm 1 and agent 2 breaks the tie in favor of arm 2. The reward vector $v$ for the principal is defined as $v_1 = 1$, $v_2 = 0.5$ and $v_3 = 0$. In each round $t$, we choose agent 1 with probability $0.4$ and agent 2 with probability $0.6$. Let this problem instance be denoted as $I_\Delta$. It can be shown that the optimal incentive is to provide an incentive of $\Delta$ to the first arm. Anything more or less will lead to a constant expected regret in each round. This rules out the possibility of discretizing the incentive space. With more refined analysis one can indeed show a linear regret.

We now formally show that any algorithm incurs a regret of $\Omega(T)$. Consider a problem instance $I_\Delta$ where $\Delta \in [0.7, 0.71]$. Observe that the expected reward for providing an incentive of $\Delta$ to the first arm is $0.6 \cdot 0.5 + 0.4 \cdot (1 - \Delta) \geq 0.416$. Next, observe that the expected reward for providing an incentive less than $\Delta$ to the first arm is at most $0.6 \cdot 0.5 + 0.4 \cdot 0 = 0.3$. This includes providing incentive to other two arms. Finally, observe that the expected reward for providing an incentive more than $\Delta$ to the first arm is at most $1 - \Delta \leq 0.3$. Hence, providing an incentive of $\Delta$ to the first arm is the best fixed incentive.

Next let us fix a deterministic algorithm `Alg`. Let $\mathcal{I} = \bigcup_{\Delta \in [0.7, 0.71]} I_\Delta$. We aim to show that $\mathbb{E}_{I \sim Unif(\mathcal{I})}[R(I, T)] = \Omega(T)$ where $R(I, T)$ is the expected regret of `Alg` on the instance $I$. W.l.o.g let us assume that in each round $t$, `Alg` chooses an incentive vector $\pi_t$ such that $(\pi_t)_1 \in [0.7, 0.71]$. Note that choosing any other type of incentive vector would lead to a constant expected regret in that round.

Now let us analyze the behavior of `Alg`. Fix a sequence of agents $j_1, j_2, \ldots, j_T$. The behavior of `Alg` can be abstractly stated as follows. In each round $t$, it chooses an incentive vector $\pi_t$ such that $(\pi_t)_1 \in [0.7, 0.71]$. An arm $i_t$ gets chosen by agent $j_t$. This can be considered as assigning the arm $i_t$ to the vector $\pi_t$. Then based on the agent numbers $j_1, \ldots, j_t$ (not their preference vectors) and arms $i_1, \ldots, i_t$, `Alg` chooses $\pi_{t+1}$. We say assignment of an arm $i_t$ to an incentive vector $\pi_t$ is consistent if one of the following holds:

- If $i_t = 1$, then for all $s < t$ such that $\pi_s \leq \pi_t$ we have $i_s = 1$.

- If $i_t = 2$, then $j_t = 2$ and for all $s < t$ such that $\pi_s \geq \pi_t$ we have $i_s > 1$.

- If $i_t = 2$, then $j_t = 3$ and for all $s < t$ such that $\pi_s \geq \pi_t$ we have $i_s > 1$.

Now consider all possible ways to consistently label the incentive vectors in all rounds. There are only finite number of possibilities for a fixed sequence of agents. Moreover, the number of possible sequence of agents is also finite. Hence, only a finite number of incentives in the range $[0.7, 0.71]$ is provided by `Alg` to the arm 1. Let this set of incentives be $\mathcal{S}$.

Now consider an instance $I_\Delta$ such that $\Delta \in [0.7, 0.71] \backslash \mathcal{S}$. Let $\mathcal{D}$ be a distribution supported on $\{1, 2\}$ such that probability of choosing 1 is $0.4$ and the probability of choosing 2 is $0.6$. Now given a filtration $\mathcal{F}_{t-1}$, $\pi_t$ gets fixed. Due to the definition of $\mathcal{S}$,

we $(\pi_t)_1 \neq \Delta$. Therefore, we have $\mathbb{E}_{j_t \sim \mathcal{D}}[U(\pi_t, j_t)|\mathcal{F}_{t-1}] \leq 0.3$. Hence, we have $\mathbb{E}[U(\pi_t, j_t)] = \mathbb{E}[\mathbb{E}[U(\pi_t, j_t)|\mathcal{F}_{t-1}]] \leq 0.3$. Hence we have $\sum_{t=1}^{T} \mathbb{E}[U(\pi_t, j_t)] \leq 0.3T$. On the other hand, for the optimal incentive vector $\pi^*$, we have $\sum_{t=1}^{T} \mathbb{E}[U(\pi^*, j_t)] \geq 0.416T$. Hence, we have $R(I_\Delta, T)$. As $\mathcal{S}$ is finite, we have $\mathbb{E}_{I \sim Unif(\mathcal{I})}[R(I, T)] = \Omega(T)$. Due to Yao's lemma, one can show that any randomized algorithm incurs a worst case expected regret of $\Omega(T)$.

## C. Lower bound for Greedy model with Single-arm incentive

In this section, we present the proof of Theorem 2.3. We begin by establishing a lower bound of $\sqrt{KT}$ for all $N \geq 3$ in Appendix C.1. Next, in Appendix C.2, we derive a lower bound of $\sqrt{KT \log(N)/\log(K)}$ for all $N \in \{K, K + 1, \ldots, 2^{(K-2)/48}\}$. This result further implies a lower bound of $\Omega(K\sqrt{T/\log(K)})$ for all $N \geq 2^{(K-2)/48}$. By combining these cases, we obtain a general lower bound of $\Omega(\min\{\sqrt{KT \log(N)/\log(K)}, K\sqrt{T/\log(K)}\})$ for all $N, K \geq 3$, thereby proving the theorem.

### C.1. $\Omega(\sqrt{KT})$ Minimax Lower Bound

For policy $A$ and instance $I$, we define the pseudo-regret

$$R_T^{A,I} = \sup_{\pi \in [0,1]^N} \mathbb{E}_{A,I} \left[ \sum_{t=1}^{T} (U(\pi, j_t) - U(\pi_t, j_t)) \right].$$

In the following, we give a lower bound for the problem with the single-arm incentive.

**Theorem C.1** (minimax lower bound). $\forall K \geq 3, N \geq 3, T > \max\{4(K-2)^3, 10(K-2)\}$, *policy* $A$, $\exists I$, $R_T^{A,I} = \Omega(\sqrt{KT})$.

*Proof.* To prove the minimax regret lower bound for adversarial selection of agent type, it suffices to prove a lower bound for the stochastic selection of agent type.

Consider any fixed $N, K, T \in \mathbb{N}$ such that $K \geq 3$, $N \geq 3$, and $T > \max\{4(K-2)^3, 10(K-2)\}$. Fix $\epsilon = \sqrt{\frac{K-2}{10T}}$. As we assume $T > 10(K-2)$, we have $\epsilon < 1/10$. To avoid clutter, we define

$$\forall i \in \{2, \ldots, K-1\}: \quad \beta_i = \frac{1}{3\left(\frac{5}{6} - \frac{i-2}{3(K-2)}\right)} - \frac{1}{3}.$$

One can easily see that $\beta_i \in [\frac{1}{15}, \frac{1}{3})$ for all $i \in \{2, \ldots, K-1\}$.

We set $v \in [0,1]^N$ as $v_1 = \frac{2}{3} + \frac{\epsilon}{3}$, $v_2 = 1$, and $v_j = 0$ for all $j \geq 3$. Each agent type $i \in [K]$ has a reward vector $\mu^i \in [0,1]^N$. We set $\mu^1 = (\frac{1}{3}, 0, \ldots, 0)$ and for the $K$-th type agent, we set $\mu_3^K = 1$, and $\mu_j^K = 0$ for all $j \in [N] \setminus \{3\}$. For every agent $i \in \{2, \ldots, K-1\}$, we set $\mu_2^i = \frac{1}{3}$, $\mu_3^i = 1 - \beta_i$, and $\mu_j^i = 0$ for all $j \in [N] \setminus \{2, 3\}$. Suppose that the agent consistently breaks the tie by choosing the arm with the smallest index, and at each round the agent type is independently sampled from a distribution, unknown to the principal.

The following gives an intuitive setting.

$$v = \left(\frac{2}{3} + \frac{\epsilon}{3}, 1, 0, 0, \ldots, 0\right)$$

$$\mu_1 = \left(\frac{1}{3}, 0, 0, 0, \ldots, 0\right)$$

$$\mu_2 = \left(0, \frac{1}{3}, 1 - \beta_2, 0, \ldots, 0\right)$$

$$\mu_3 = \left(0, \frac{1}{3}, 1 - \beta_3, 0, \ldots, 0\right)$$

$$\vdots$$

$$\mu_{K-1} = \left(0, \frac{1}{3}, 1 - \beta_{K-1}, 0, \ldots, 0\right)$$

$$\mu_K = (0, 0, 1, 0, \ldots, 0).$$

We consider a class of instances, denoted by $\mathcal{I}$ with fixed $v, \{\mu_i\}_{i \in [K]}$ and the breaking rule given above, but allow all possible probability distributions over the agent type with $p_1 = \frac{1}{2}$, i.e., the probability of selecting the first type of agent is equal to $\frac{1}{2}$. Let $\Pi = \{\pi_i\}_{i=1}^{K-1}$ where $\pi_1 = (0, \ldots, 0)$ and for each $i \in \{2, \ldots, K-1\}$, $\pi_i = (0, \frac{2}{3} - \beta_i, 0, \ldots, 0)$, i.e., the second coordinate takes the value of $\frac{2}{3} - \beta_i$, which is the minimum cost to incentivize $i$-th type of agent to play arm 2, and all others are zero. Then, we consider a policy set $\mathcal{A}$, which maps from history to $\Pi$, i.e., at each round, the principal only offers an incentive among $\Pi$. We only need to lower bound $\inf_{A \in \mathcal{A}} \sup_{I \in \mathcal{I}} R_T^{A,I}$ since

$$\inf_A \sup_I R_T^{A,I} \geq \inf_A \sup_{I \in \mathcal{I}} R_T^{A,I} = \inf_{A \in \mathcal{A}} \sup_{I \in \mathcal{I}} R_T^{A,I},$$

where the equality holds because for all $I \in \mathcal{I}$ and all $A \notin \mathcal{A}$, if policy $A$ offers a single-value incentive $\pi \notin \Pi$ at a certain round, then one can always alternatively propose an incentive $\pi_i = (\pi_{i,1}, \ldots, \pi_{i,K}) \in \Pi$ (for some $i \in [K-1]$) incurs an instantaneous regret no worse than $\pi$.

To prove the above claim, it suffices to consider two types single-arm incentive which has positive value on either arm 1 or arm 2. For any single-arm incentive $\pi \notin \Pi$ which has positive value on arm 1, if the value is strictly smaller than $1 - \beta_{K-1}$, then it functions the same as $(0, \ldots, 0)$, in the sense that their expected utilities are the same. For any other value larger than or equal to $1 - \beta_{K-1}$, incentivizing any agent $i \in [K]$ to play arm 1 gives utility at most $\frac{2+\epsilon}{3} - (1 - \beta_{K-1})$, which is negative. Notice that $\frac{2+\epsilon}{3} - (1 - \beta_{K-1}) < 0$ can be verified by using $\epsilon = \sqrt{\frac{K-2}{10T}}$ and $T > 4(K-2)^3$. However, $p_1 = \frac{1}{2}$ implies that incentive $(0, \ldots, 0)$ yields the expected utility $\frac{1}{3} + \frac{\epsilon}{6}$. On the other hand, for any single-arm incentive $\pi \notin \Pi$ which has positive value on arm 2, if the value is strictly smaller than $\frac{2}{3} - \beta_{K-1}$, then it again functions the same as $(0, \ldots, 0)$. If the value is larger than or equal to $\frac{2}{3} - \beta_{K-1}$, then there must exist $i \in [K-1]$ such that the expected utility of $\pi_i$ is no worse than that of $\pi$.

Now, we construct two instances $I, I' \in \mathcal{I}$.

**Construction of Instance $I$.** For instance $I$, the agent type is sampled from a distribution $p = (p_1, \ldots, p_K)$ where

$$p_i = \begin{cases} \frac{1}{2} & \text{if } i = 1, \\ \frac{1}{3(K-2)} & \text{if } i \in \{2, \ldots, K-1\}, \\ \frac{1}{6} & \text{if } i = K. \end{cases}$$

We treat each incentive in $\Pi$ as an arm, and $|\Pi| = K - 1$. Let $r(\pi_i)$ be the expected utility of arm (incentive) $\pi_i$. For shorthand, we write $r_i := r(\pi_i)$. For instance $I$, the expected utility vector of arms is denoted by $(r_1, \ldots, r_{K-1})$. Each arm follows a Bernoulli-type distribution. Arm 1 corresponds to the incentive $(0, 0, \ldots, 0)$ and takes value $\frac{2}{3} + \frac{\epsilon}{3}$ with probability $p_1 = \frac{1}{2}$, and takes 0 with equal probability, which implies that its expected utility is

$$r_1 = v_1 p_1 = \frac{1}{2}\left(\frac{2}{3} + \frac{\epsilon}{3}\right) = \frac{1}{3} + \frac{\epsilon}{6}.$$

Similarly, arm (incentive) $\pi_2$ takes the utility of $\frac{2}{5}$ with probability $1 - p_K = \frac{5}{6}$ and takes 0 with remaining probability, because it takes the value of $\frac{2}{5}$ as long as the agent $K$ does not show up at that certain round, and thus the expected utility is $r_2 = \frac{1}{3}$. Moreover, one can verify that for each arm $\pi_i$ with $i \in \{3, \ldots, K-1\}$, the utility takes value of $1 - \pi_{i,2} = \frac{1}{3} + \beta_i$ with probability $1 - p_K - \sum_{z \in \{2, \ldots, i-1\}} p_z$, and takes 0 with remaining probability. Thus the expected utility is

$$r_i = \left(1 - p_K - \sum_{z \in \{2, \ldots, i-1\}} p_z\right)(1 - \pi_{i,2}) = \left(\frac{5}{6} - \frac{i-2}{3(K-2)}\right)\left(\frac{1}{3} + \beta_i\right) = \frac{1}{3}.$$

Thus, arm $\pi_1$ is uniquely optimal, and the arm gaps for all others are the same, denoted by

$$\Delta := \frac{\epsilon}{6}.$$

**Construction of Instance $I'$.** Then, we fix an arbitrary policy $A \in \mathcal{A}$ and construct another instance $I'$ by modifying $p$ in instance $I$ to $p'$. Let $z = \arg\min_{j \in \{2,\ldots,K-1\}} \mathbb{E}_{A,I_1}[T_j]$ where $T_j$ is the number of plays of arm $j$ across $T$ rounds. Then, we set the probability distribution $p' = (p'_1, \ldots, p'_K)$ by considering two cases of $z$. For shorthand, let

$$l_{z,\epsilon} = \epsilon \left( \frac{5}{6} - \frac{z-2}{3(K-2)} \right) \in \left( \frac{\epsilon}{2}, \frac{5\epsilon}{6} \right].$$

**Case 1: $z \geq 3$.** In this case, we set

$$p'_i = \begin{cases} \frac{1}{2} & i = 1, \\ \frac{1}{3(K-2)} & i \in \{2, \ldots, K-1\} - \{z-1, z\}, \\ \frac{1}{3(K-2)} - l_{z,\epsilon} & i = z-1, \\ \frac{1}{3(K-2)} + l_{z,\epsilon} & i = z, \\ \frac{1}{6} & i = K. \end{cases}$$

**Case 2: $z = 2$.** In this case, we set

$$p'_i = \begin{cases} \frac{1}{2} & i = 1, \\ \frac{1}{3(K-2)} & i \in \{3, \ldots, K-1\}, \\ \frac{1}{3(K-2)} + l_{z,\epsilon} & i = 2, \\ \frac{1}{6} - l_{z,\epsilon} & i = K. \end{cases}$$

Notice that one can make a sanity check for all $z$

$$\frac{1}{3(K-2)} - l_{z,\epsilon} \geq \frac{1}{3(K-2)} - \frac{5\epsilon}{6} = \frac{1}{3(K-2)} - \frac{5}{6}\sqrt{\frac{K-2}{10T}} > 0,$$

where the last inequality holds due to the assumption $T > (K-2)^3$. Moreover, $\frac{1}{6} - l_{z,\epsilon} \geq \frac{1-5\epsilon}{6} > 0$ as $\epsilon < 1/10$. Therefore, $p'$ is a valid distribution for both cases.

In Case 1 with $z \geq 3$, the expected utility of arm $z$ in instance $I_2$ is

$$r'_z = \left( 1 - p_K - \sum_{j \in \{2, \ldots, i-1\}} p_j + l_{z,\epsilon} \right) \left( \frac{1}{3} + \beta_i \right) = r_z + 2\Delta.$$

Similarly, in Case 2 with $z = 2$, we also have $r'_z = r_z + 2\Delta$. Thus, the expected utility vector in instance $I_2$ is

$$(r_1, \ldots, r_z + 2\Delta, \ldots, r_{K-1}).$$

In this case, arm $z$ becomes the unique optimal arm. For instances $I_1, I_2$, the utility distribution of each arm $i \in [K-1] - \{z\}$ remains unchanged, but only the one of arm $z$ slightly changes.

**Proof of Lower Bound.** Let $(D_1, \ldots, D_{K-1})$ and $(D'_1, \ldots, D'_{K-1})$ be the utility distributions over arm $\Pi$ in instances $I, I'$, respectively. For shorthand, if $z = 2$, let $q_z = \frac{5}{6}$, and if $z \in \{3, \ldots, K-1\}$, let $q_z = \frac{5}{6} - \sum_{j \in \{2, \ldots, i-1\}} p_j$. We use $\mathrm{KL}(P, Q)$ to denote the KL divergence between two distributions $P, Q$. We define an event $\mathcal{E} := \left\{ T_1 \leq \frac{T}{2} \right\}$ and its complement as $\mathcal{E}^c$. Then, by a standard lower-bound argument used in MAB, we have

$$\begin{aligned} R_T^{A,I} + R_T^{A,I'} &\geq \frac{T\Delta}{2} \left( \mathbb{P}_{A,I}(\mathcal{E}) + \mathbb{P}_{A,I'}(\mathcal{E}^c) \right) \\ &\geq \frac{T\Delta}{2} \left( 1 - \max_{\mathcal{E}} |\mathbb{P}_{A,I}(\mathcal{E}) - \mathbb{P}_{A,I'}(\mathcal{E}^c)| \right) \\ &\geq \frac{T\Delta}{2} \left( 1 - \sqrt{\frac{1}{2}\mathrm{KL}\left( \mathbb{P}_{A,I} || \mathbb{P}_{A,I'} \right)} \right), \end{aligned}$$

where the second inequality uses $\mathbb{P}_{A,I}(\mathcal{E}) + \mathbb{P}_{A,I'}(\mathcal{E}^c) = \mathbb{P}_{A,I}(\mathcal{E}) + 1 - \mathbb{P}_{A,I'}(\mathcal{E}) \geq 1 - |\mathbb{P}_{A,I}(\mathcal{E}) - \mathbb{P}_{A,I'}(\mathcal{E})| \geq 1 - \max_{\mathcal{E}} |\mathbb{P}_{A,I}(\mathcal{E}) - \mathbb{P}_{A,I'}(\mathcal{E})|$, and the third inequality follow from Pinsker's inequality.

The definition of $z$ implies that $\mathbb{E}_{A,I}[T_z] \leq T/(K-2)$. By the divergence decomposition lemma, we have

$$\mathrm{KL}\left(\mathbb{P}_{A,I}||\mathbb{P}_{A,I'}\right) = \mathbb{E}_{A,I}[T_z]\mathrm{KL}(D_z||D'_z) \leq \frac{180T\Delta^2}{K-2}.$$

Therefore, we have

$$R_T^{A,I} + R_T^{A,I'} \geq \frac{T\Delta}{2}\left(1 - \Delta\sqrt{\frac{90T}{K-2}}\right).$$

Recall that $\Delta = \epsilon/6$. Finally, as $\epsilon = \sqrt{\frac{K-2}{10T}}$, we have $\Delta = \frac{\epsilon}{6} = \sqrt{\frac{K-2}{360T}}$, which implies that $1 - \Delta\sqrt{\frac{90T}{K-2}} \geq \frac{1}{2}$. Thus,

$$R_T^{A,I} + R_T^{A,I'} = \Omega(\sqrt{KT}),$$

which suffices to give the lower bound.

**Bounding** $\mathrm{KL}(D_z||D'_z)$**.** In instances $I, I'$, arm $z$ take the same positive value with probability $q_z$ and $q_z + l_{z,\epsilon}$ respectively, and zero with remaining probabilities. We bound

$$\mathrm{KL}(D_z||D'_z) = q_z \log\left(\frac{q_z}{q_z + l_{z,\epsilon}}\right) + (1 - q_z)\log\left(\frac{1 - q_z}{1 - q_z - l_{z,\epsilon}}\right).$$

By using the fact that $\log(1 + x) \geq x - x^2/2$ for any $x \geq 0$, we have

$$q_z \log\left(\frac{q_z}{q_z + l_{z,\epsilon}}\right) = -q_z \log\left(1 + \frac{l_{z,\epsilon}}{q_z}\right) \leq -q_z\left(\frac{l_{z,\epsilon}}{q_z} - \frac{l_{z,\epsilon}^2}{2q_z^2}\right) = l_{z,\epsilon} - \frac{l_{z,\epsilon}^2}{2q_z}.$$

The second term is bounded by

$$\begin{aligned}
(1 - q_z)\log\left(\frac{1 - q_z}{1 - q_z - l_{z,\epsilon}}\right) &= -(1 - q_z)\log\left(1 - \frac{l_{z,\epsilon}}{1 - q_z}\right) \\
&\leq (1 - q_z)\left(\frac{l_{z,\epsilon}}{1 - q_z} + \frac{1}{2}\left(\frac{l_{z,\epsilon}}{1 - q_z}\right)^2 + \frac{2}{3}\left(\frac{l_{z,\epsilon}}{1 - q_z}\right)^3\right) \\
&= l_{z,\epsilon} + \frac{l_{z,\epsilon}^2}{2(1 - q_z)} + \frac{2}{3}\frac{l_{z,\epsilon}^3}{(1 - q_z)^2},
\end{aligned}$$

where the second inequality uses series expansion and the fact that $\frac{l_{z,\epsilon}}{1 - q_z} \leq 5\epsilon < 1/2$ as $q_z \leq \frac{5}{6}$ and $\epsilon < 1/10$.

Combining both and using $\Delta = \epsilon/6$, we have

$$\mathrm{KL}(D_z||D'_z) \leq \frac{l_{z,\epsilon}^2}{2(1 - q_z)} - \frac{l_{z,\epsilon}^2}{2q_z} + \frac{2}{3}\frac{l_{z,\epsilon}^3}{(1 - q_z)^2} \leq 5\epsilon^2 = 180\Delta^2.$$

Thus, the proof completes. $\qquad\qquad\qquad\qquad\qquad\qquad\qquad\qquad\qquad\qquad\qquad\qquad\qquad\qquad\qquad$ $\square$

## C.2. $\Omega\left(\sqrt{KT\log(N)/\log(K)}\right)$ worst case lower bound

In this section, we construct an instance with $K$ agents and $N$ arms and establish a worst-case lower bound of $\Omega\left(\sqrt{\frac{KT\log N}{\log K}}\right)$ for all $N \in \{K, K+1, \ldots, 2^{(K-2)/48}\}$. This also implies a worst-case lower bound of $\Omega\left(K\sqrt{\frac{T}{\log K}}\right)$ for all $N \geq 2^{(K-2)/48}$.

Let $K_0 := K - 2$, and define $M := \frac{\log(N-1)}{\log(K-2)}$. Let $N_0 := \left(\frac{K_0}{M}\right)^M + 1$. Note that $\log(N_0 - 1) = M \log(K_0/M) < M \log(K_0) = \log(N - 1)$, which implies $N_0 < N$.

For simplicity, we assume that both $M$ and $\frac{K_0}{M}$ are integers. We also assume $K \geq 1000$. Fix an $\varepsilon > 0$ to be specified later in the proof.

Let $d_i = (i - 1) \cdot \frac{K_0}{M}$ for all $j \in [M]$. Let $\mathcal{X} := \{x \in \{0,1\}^{K_0} : \forall i \in [M] \sum_{j=1}^{K_0/M} x_{d_i+j} = 1\}$. Now consider a bijective mapping $f : \mathcal{X} \mapsto [N_0 - 1]$. First we define the reward vector $v$ for the principal. We have $v_i := 0.5 + \frac{1}{T}$ for each $i \in [N_0 - 1]$ and $v_i := 0$ for all $i \in \{N_0, N_0 + 1, \ldots, N\}$.

Next we define the preference vector $\mu^j$ for an agent $j \in [K_0]$. For each $x \in \mathcal{X}$, we define $\mu^j_{f(x)} := 1 - \frac{1}{T}$ if $x_j = 1$ and $\mu^j_{f(x)} = 0$ otherwise. We define $\mu^j_{N_0} := 1$ and $\mu^j_i := 0$ for all $i \in [N] \setminus [N_0]$. Next we define the preference vector $\mu^{K-1}$ for the agent $K - 1$. We have $\mu^{K-1}_i := \frac{1}{T}$ if $i = N_0$ and $\mu^K_i := 0$ otherwise. Finally we define the preference vector $\mu^K$ for the agent $K$. We have $\mu^K_i := 1$ if $i = N_0$ and $\mu^K_i := 0$ otherwise. In case of tie, an agent prefers an arm in $[N_0 - 1]$ over the arm $N_0$. Let $\Pi = \{\pi^x\}_{x \in \mathcal{X}}$ be a set of incentive vectors.

For all $x \in \mathcal{X}$, we have $\pi^x_i = \frac{1}{T}$ if $i = f(x)$ and $\pi^x_i = 0$ otherwise. Now observe that for any incentive vector $\pi \in [0,1]^N$, there always exists a vector $\pi^x \in \Pi$ such that the regret incurred by $\pi^x$ is at most the regret incurred by $\pi$. Hence, we assume that any algorithm chooses one of the incentive vectors from $\Pi$. Let us fix one such algorithm, say Alg and assume that Alg is deterministic. Now we show that algorithm Alg incurs a regret of $\Omega(\sqrt{KT \log(N)/\log(K)})$. One can easily extend the result to randomized algorithms using Yao's lemma.

Let $\tilde{\mathcal{X}} := \{x \in \{0,1\}^{K_0} : \forall i \in [M] \sum_{j=1}^{K_0/M} x_{d_i+j} \leq 1\}$. We define a mapping $p : \tilde{\mathcal{X}} \times [K] \mapsto [0,1]$ as follows. First, we define $p(x, K) = \frac{1}{4}$ for any $x \in \tilde{\mathcal{X}}$. Next, we define $p(x, K) = \frac{1}{4} - \frac{\varepsilon}{M} \cdot ||x||_1$ for any $x \in \tilde{\mathcal{X}}$. Next for any $x \in \tilde{\mathcal{X}}$ and $j \in [K_0]$, we have $p(x, j) = \frac{1}{2K_0} + x_j \cdot \frac{\varepsilon}{M}$.

We now describe an input instance $I_x$, where $x \in \tilde{\mathcal{X}}$. In the instance $I_x$, in each round an agent $j$ arrives with probability $p(x, j)$. If $x$, then observe that the expected reward of playing an incentive vector $\pi^z$ under the instance $I_x$ is $r(x, z) := \frac{1}{4} \cdot \frac{1}{2} + \sum_{j=1}^{K_0} p(x, j) \cdot \mathbb{1}\{z_j = 1\} \cdot \frac{1}{2} = \frac{1}{8} + \frac{M}{4K_0} + \frac{\varepsilon}{2M} \cdot \sum_{j=1}^{K_0} \mathbb{1}\{z_j = 1, x_j = 1\}$, where $z \in \mathcal{X}$. If $x \in \mathcal{X}$, then observe that $\pi^x$ is the optimal incentive vector for the instance $I_x$ and the expected reward is $\mu^* := \frac{1}{8} + \frac{M}{4K_0} + \frac{\varepsilon}{2}$. Let $\mathcal{I} = \bigcup_{x \in \mathcal{X}} I_x$ be the set of input instances that we analyze our regret on.

At each round $t$, suppose Alg selects $\pi^{z_t} \in \Pi$. For the instance $I_x$ with $x \in \mathcal{X}$, the cumulative regret after $T$ rounds, $R_x(T)$, is defined as:

$$R_x(T) = T \cdot \mu^* - \mathbb{E}_{I_x}\left[\sum_{t=1}^{T} r(x, z_t)\right] = \frac{\varepsilon T}{2} - \frac{\varepsilon}{2M} \sum_{t=1}^{T} \sum_{j=1}^{K_0} \mathbb{P}_{I_x}[(z_t)_j = 1, x_j = 1],$$

where $\mathbb{P}_{I_x}$ is probability law under the instance $I_x$

Recall that $d_i = (i - 1) \cdot \frac{K_0}{M}$ for all $i \in [M]$. For any instance $I_x$ with $x \in \mathcal{X}$, the regret can be written as $R_x(T) = \sum_{i=1}^{M} R_{x,i}(T)$, where

$$R_{x,i}(T) = \frac{\varepsilon T}{2M} - \frac{\varepsilon}{2M} \sum_{t=1}^{T} \sum_{j=1}^{K_0/M} \mathbb{P}_{I_x}[(z_t)_{d_i+j} = 1, x_{d_i+j} = 1].$$

Fix an index $i \in [M]$. Let $\mathcal{X}^{(i)} = \{x \in \{0,1\}^{K_0} : \forall j \in [M] \setminus \{i\} \sum_{s=1}^{M} x_{d_j+s} = 1, \sum_{s=1}^{M} x_{d_i+s} = 0\}$. For any vector $x \in \mathcal{X}^{(i)}$, let $x^{(j)}$ denote a vector in $\mathcal{X}$ such that $x^{(j)}_{d_i+j} = 1$ and $x^{(j)}_s = x$ for all $s \in [K_0] \setminus \{d_i + j\}$. We claim that for any $x \in \mathcal{X}^{(i)}$, there exists a set $\mathcal{S}_x \subseteq \left[\frac{K_0}{M}\right]$ of size at least $\frac{K_0}{3M}$ such that for each $j \in \mathcal{S}_x$, we have $R_{x^{(j)},i}(T) \geq c \cdot \sqrt{\frac{K_0}{M} \cdot T}$, where $c$ is an absolute constant.

Before proving the claim, we first show that if it holds for any index $i \in [M]$, then we have $\mathbb{E}_{I_x \sim Unif(\mathcal{I})}[R_x(T)] \geq c' \cdot \sqrt{MK_0 T} = c' \cdot \sqrt{(K - 2) \cdot T \cdot \frac{\log(N-1)}{\log(K-2)}}$ where $c'$ is an absolute constant. As $R_x(T) = \sum_{i=1}^{M} R_{x,i}(T)$, it suffices to

show that $\mathbb{E}_{I_x \sim Unif(\mathcal{I})}[R_{x,i}(T)] \geq c' \cdot \sqrt{\frac{K_0}{M} \cdot T}$ for all $i \in [M]$. Fix an index $i \in [M]$. Now we have the following:

$$
\begin{aligned}
\mathbb{E}_{I_{x'} \sim Unif(\mathcal{I})}[R_{x',i}(T)] &= \frac{1}{(K_0/M)^M} \sum_{x \in \mathcal{X}^{(i)}} \sum_{j=1}^{K_0/M} R_{x^{(j)},i}(T) \\
&\geq \frac{1}{(K_0/M)^M} \sum_{x \in \mathcal{X}^{(i)}} \sum_{j \in \mathcal{S}_x} R_{x^{(j)},i}(T) \\
&\geq \frac{c}{(K_0/M)^M} \sum_{x \in \mathcal{X}^{(i)}} \sum_{j \in \mathcal{S}_x} \sqrt{\frac{K_0}{M} \cdot T} \\
&\geq \frac{c}{(K_0/M)^M} \sum_{x \in \mathcal{X}^{(i)}} \frac{K_0}{3M} \cdot \sqrt{\frac{K_0}{M} \cdot T} \\
&= \frac{c}{(K_0/M)^M} \cdot (K_0/M)^{M-1} \cdot \frac{K_0}{3M} \cdot \sqrt{\frac{K_0}{M} \cdot T} \\
&= \frac{c}{3} \cdot \sqrt{\frac{K_0}{M} \cdot T}
\end{aligned}
$$

We now prove the claim for a fixed index $i \in [M]$ and a fixed vector $x \in \mathcal{X}^{(i)}$, using the chain rule from Lemma A.2 in our analysis.

For an instance $I_{x^{(j)}}$, let $f_j(a_1, \ldots, a_T)$ denote the joint PMF over the sequence of binary decisions by the agent in each round under the probability distribution $\mathbb{P}_{I_{x^{(j)}}}$. The sample space is $\Omega = \{0, 1\}^T$, where $0$ indicates that the agent selects arm $N_0$, and $1$ indicates that the agent selects the arm incentivized by Alg. This is a valid sample space, as the agent chooses either the arm incentivized by Alg or arm $N_0$ in each round. Similarly, for the alternate instance $I_x$, let $f_0(a_1, \ldots, a_T)$ denote the joint PMF of the binary decisions under the distribution $\mathbb{P}_{I_x}$.

First, observe that the instances $I_{x^{(j)}}$ and $I_x$ differ only at agent $d_i + j$. For each $\omega \in \Omega$, let $\pi^{z_1,\omega}, \pi^{z_2,\omega}, \ldots, \pi^{z_T,\omega}$ denote the sequence of incentive vectors chosen by Alg on $\omega$. Conditioning on the outcomes $X_1 = \omega_1, X_2 = \omega_2, \ldots, X_{t-1} = \omega_{t-1}$, we have $X_t \sim \text{Ber}(\tilde{\mu}_j)$ under instance $I_{x^{(j)}}$, and $X_t \sim \text{Ber}(\tilde{\mu}_0)$ under instance $I_x$, where $\tilde{\mu}_j - \tilde{\mu}_0 = \frac{\varepsilon}{M} \cdot (z_{t,\omega})_{d_i+j}$. Define $T_j = \sum_{t=1}^T (z_t)_{d_i+j}$ as a random variable, and for each $\omega \in \Omega$, let $T_{j,\omega} = \sum_{t=1}^T (z_{t,\omega})_{d_i+j}$, which is a fixed value. Now we have the following:

$$
\begin{aligned}
\text{KL}(f_0, f_j) &= \sum_{\omega \in \Omega} f_0(\omega) \left( \text{KL}(f_0(X_1), f_j(X_1)) + \sum_{t=2}^T \text{KL}(f_0(X_t | X_{-t} = \omega_{-t}), f_j(X_t | X_{-t} = \omega_{-t})) \right) \\
&\leq \frac{c_0 \varepsilon^2}{M^2} \sum_{\omega \in \Omega} f_0(\omega) \sum_{t=1}^T (z_{t,\omega})_{d_i+j} \qquad\qquad \text{(due to Lemma A.1)} \\
&= \frac{c_0 \varepsilon^2}{M^2} \sum_{\omega \in \Omega} f_0(\omega) T_{j,\omega} \\
&= \frac{c_0 \varepsilon^2}{M^2} \cdot \mathbb{E}_{I_x}[T_j]
\end{aligned}
$$

where $c_0$ is some absolute constant.

Observe that $\sum_{j=1}^{K_0/M} \mathbb{E}_{I_x}[T_j] = T$. Therefore, there exists a subset $\mathcal{S}_x \subseteq \left[\frac{K_0}{M}\right]$ of size at least $\frac{K_0}{3M}$ such that for each $j \in \mathcal{S}_x$, we have $\mathbb{E}_{I_x}[T_j] \leq \frac{3MT}{K_0}$. Fix $\varepsilon = \sqrt{\frac{MK_0}{25c_0 T}}$. Then for each $j \in \mathcal{S}_x$, we have $\text{KL}(f_0, f_j) \leq \frac{3c_0 \varepsilon^2 T}{MK_0} = \frac{3}{25}$.

Fix an index $j \in \mathcal{S}_x$, and let $A_j$ denote the event that $T_j \leq \frac{12MT}{K_0}$. By Markov's inequality, we have $\mathbb{P}_{I_x}(A_j) \geq \frac{3}{4}$. Then, by Pinsker's inequality, we obtain the following:

$$
\mathbb{P}_{I_{x^{(j)}}}(A_j) \geq \mathbb{P}_{I_x}(A_j) - \sqrt{\frac{\text{KL}(f_0, f_j)}{2}}
$$

$$\geq \frac{3}{4} - \sqrt{\frac{3}{50}}$$

$$> \frac{1}{2}$$

Using the inequality above, we have $\mathbb{E}_{I_{x(j)}}[T_j] \leq T \cdot \mathbb{P}_{I_{x(j)}}(A_j^c) + \frac{12MT}{K_0} \leq \frac{3T}{4}$, since $K_0/M \geq 48$. Now we have the following:

$$
\begin{aligned}
R_{x^{(j)},i}(T) &= \frac{\varepsilon T}{2M} - \frac{\varepsilon}{2M} \sum_{t=1}^{T} \sum_{s=1}^{K_0/M} \mathbb{P}_{I_{x(j)}}[(z_t)_{d_i+s} = 1, x_{d_i+s}^{(j)} = 1] \\
&= \frac{\varepsilon T}{2M} - \frac{\varepsilon}{2M} \mathbb{E}_{I_{x(j)}}[\sum_{t=1}^{T} z_t[d_i + j]] \\
&= \frac{\varepsilon T}{2M} - \frac{\varepsilon}{2M} \mathbb{E}_{I_{x(j)}}[T_j] \\
&\geq \frac{\varepsilon T}{2M} - \frac{3\varepsilon T}{8M} \\
&= \frac{1}{40} \cdot \sqrt{\frac{K_0 T}{c_0 M}}
\end{aligned}
$$

**Remark:** All the calculations and probability expressions in this section are valid for $T \geq \text{poly}(K)$.

## D. Linear Bandit based approach for Principal-agent problem with single arm incentive.

We make a natural assumption on any agent's tie braking rule. If there is a tie among a set of arms $\mathcal{I} \subseteq [N]$ and an arm $i_\star \in \mathcal{I}$ gets chosen by an agent $j$, then for a tie among a set of arms $\mathcal{I}' \subseteq \mathcal{I}$ such that $i_\star \in \mathcal{I}'$ the agent $j$ again chooses the arm $i_\star$. Towards the end of the section we show a way to circumvent this assumption. For simplicity of presentation, we also assume that each preference vector has at least two coordinates with different values.

First, we discretize the space of incentive vectors $\mathcal{D} = \{x \in [0,1]^N : |\text{supp}(x)| \leq 1\}$ as follows. We initialize an empty set $\Pi$. For each arm $i \in [N]$ and agent $j \in [K]$, let $\pi^{i,j}$ be an incentive vector defined by $\pi_s^{i,j} = 0$ for $s \neq i$ and $\pi_i^{i,j} = \max_{k \in [N]} \mu_k^j - \mu_i^j$. We then add $\pi^{i,j}$ to $\Pi$. For all $i \in [N]$, we define $\pi^{i,K+1} := (0,0,\ldots,0)$, noting that this incentive vector is already included in $\Pi$.

Now, we define $\Delta := \min_{i \in [N]} \min_{j_1,j_2 \in [K]:\pi_i^{i,j_1} \neq \pi_i^{i,j_2}} |\pi_i^{i,j_1} - \pi_i^{i,j_2}|$ and set $\varepsilon_T := \min\left\{\frac{\Delta}{2}, \frac{1}{2T}\right\}$. For each $i \in [N]$ and $j \in [K+1]$, let $\tilde{\pi}^{i,j}$ be an incentive vector defined by $\tilde{\pi}_s^{i,j} = 0$ for $s \neq i$ and $\tilde{\pi}_i^{i,j} = \pi_i^{i,j} + \varepsilon_T$. We then add $\tilde{\pi}^{i,j}$ to the set $\Pi$. We now claim that the following holds for any sequence of agents $j_1, j_2, \ldots, j_T$:

$$\max_{\pi \in \Pi} \sum_{t=1}^{T} U(\pi, j_t) \geq \left(\sup_{\pi \in \mathcal{D}} \sum_{t=1}^{T} U(\pi, j_t)\right) - 1. \tag{3}$$

Consider $\hat{\pi} \in \mathcal{D}$ such that $\sum_{t=1}^{T} U(\hat{\pi}, j_t) \geq \left(\sup_{\pi \in \mathcal{D}} \sum_{t=1}^{T} U(\pi, j_t)\right) - \varepsilon_T$. Let $\hat{i} = \arg\max_{i \in [N]} \hat{\pi}_i$. If there exists an index $j \in [K+1]$ such that $\hat{\pi}_{\hat{i}} = \pi_{\hat{i}}^{\hat{i},j}$, then the inequality (3) clearly holds. Otherwise, let $\hat{j} = \arg\max_{j \in [K+1]:\pi_{\hat{i}}^{\hat{i},j} < \hat{\pi}_{\hat{i}}} \pi_{\hat{i}}^{\hat{i},j}$. Observe that $b(\hat{\pi}, j_t) = b(\tilde{\pi}^{\hat{i},\hat{j}}, j_t)$ for all $t \in [T]$. If $\tilde{\pi}_{\hat{i}}^{\hat{i},\hat{j}} \leq \hat{\pi}_{\hat{i}}$, then $\sum_{t=1}^{T} U(\tilde{\pi}^{\hat{i},\hat{j}}, j_t) \geq \sum_{t=1}^{T} U(\hat{\pi}, j_t) \geq \left(\sup_{\pi \in \mathcal{D}} \sum_{t=1}^{T} U(\pi, j_t)\right) - \varepsilon_T$. If $\tilde{\pi}_{\hat{i}}^{\hat{i},\hat{j}} > \hat{\pi}_{\hat{i}}$, then $U(\tilde{\pi}^{\hat{i},\hat{j}}, j_t) \geq U(\hat{\pi}, j_t) - \varepsilon_T$ as $\pi_{\hat{i}}^{\hat{i},\hat{j}} < \hat{\pi}_{\hat{i}} < \tilde{\pi}_{\hat{i}}^{\hat{i},\hat{j}} = \pi_{\hat{i}}^{\hat{i},\hat{j}} + \varepsilon_T$. Hence, $\sum_{t=1}^{T} U(\tilde{\pi}^{\hat{i},\hat{j}}, j_t) \geq \sum_{t=1}^{T} U(\hat{\pi}, j_t) - T \cdot \varepsilon_T \geq \left(\sup_{\pi \in \mathcal{D}} \sum_{t=1}^{T} U(\pi, j_t)\right) - 1$. Thus, the inequality (3) holds in this case as well.

The rest of the proof is identical to the main body. Nevertheless, we include it for completeness. Let $\mathbf{0}$ denote the incentive vector $(0,0,\ldots,0)$. Let us define a mapping $h : \Pi \to \{0,1\}^K$. Consider $\pi \in \Pi \setminus \{\mathbf{0}\}$. Let $a(\pi) := \arg\max_{i \in [N]} \pi_i$. For any $j \in [K]$, we have $(h(\pi))_j = 1$ if $b(\pi, j) = a(\pi)$ otherwise we have $(h(\pi))_j = 0$. We now construct a new

set $\widehat{\Pi}$ as follows. Consider a vector $s \in \{0, 1\}^K$. Let $\Pi_s := \{\pi \in \Pi \setminus \{\mathbf{0}\} : h(\pi) = s\}$. If $|\Pi_s| = 1$, then add the incentive vector in $\Pi_s$ to $\widehat{\Pi}$. If $|\Pi_s| > 1$, let $\hat{\pi} = \arg\max_{\pi \in \Pi_s} v_{a(\pi)} - \pi_{a(\pi)}$. Observe that for any $j \in [K]$, if $s_j = 1$ then $U(\hat{\pi}, j) = v_{a(\hat{\pi})} - \hat{\pi}_{a(\hat{\pi})} = \max_{\pi \in \Pi_s} v_{a(\pi)} - \pi_{a(\pi)} = \max_{\pi \in \Pi_s} U(\pi, j)$. On the other hand if $s_j = 0$, then $b(\hat{\pi}, j) = b(\pi, j)$ and $\hat{\pi}_{b(\hat{\pi}, j)} = \hat{\pi}_{b(\pi, j)} = 0$ for any $\pi \in \Pi_s$. Therefore we have $U(\hat{\pi}, j) = U(\pi, j)$ for any $\pi \in \Pi_s$. We now add $\hat{\pi}$ to $\widehat{\Pi}$. We also add $\mathbf{0}$ to $\widehat{\Pi}$. We have $|\widehat{\Pi}| \leq \min\{2KN + N, 2^K\} + 1$. Now observe that for any sequence of agents $j_1, j_2, \dots, j_T$ we have:

$$\max_{\pi \in \widehat{\Pi}} \sum_{t=1}^{T} U(\pi, j_t) \geq \left( \sup_{\pi \in \mathcal{D}} \sum_{t=1}^{T} U(\pi, j_t) \right) - 1 \tag{4}$$

Hence, it suffices to focus on the set of incentive vectors $\widehat{\Pi}$ for our regret minimization problem. Now we show a reduction of this problem to an adversarial linear bandit problem. First we construct a set $\widehat{\mathcal{Z}} \subset \mathbb{R}^K$ as follows. For each $\pi \in \widehat{\Pi}$, we add $z^\pi$ to $\widehat{\mathcal{Z}}$ where $(z^\pi)_j = U(\pi, j)$ for all $j \in [K]$. Next we define the reward vector $y_t \in \mathbb{R}^K$. If agent $j_t$ has arrived in round $t$, we set $(y_t)_j = 1$ if $j = j_t$ and zero otherwise. Now observe that for any $\pi \in \widehat{\Pi}$, $U(\pi, j_t) = \langle z^\pi, y_t \rangle$. Hence, our pseudo-regret $R_T$ is equal to the adversarial linear bandit pseudo-regret:

$$R_T = \max_{z \in \widehat{\mathcal{Z}}} \mathbb{E} \left[ \sum_{t=1}^{T} \langle z, y_t \rangle - \sum_{t=1}^{T} \langle z_t, y_t \rangle \right].$$

By using EXP3 for linear bandits, we get a regret upper bound of $\mathcal{O}\left( \min \left\{ \sqrt{KT \log(KN)}, K\sqrt{T} \right\} \right)$ for our regret minimization problem.

Now let us consider the case when the natural assumption does not hold. In that case, we instead on the set of incentive vectors $\Pi$ for our regret minimization problem. Now we show a reduction of this problem to an adversarial linear bandit problem. First, we construct a set $\mathcal{Z} \subset \mathbb{R}^K$ as follows: for each $\pi \in \Pi$, we add $z^\pi$ to $\mathcal{Z}$, where $(z^\pi)_j = U(\pi, j)$ for all $j \in [K]$. Next, we define the reward vector $y_t \in \mathbb{R}^K$. If agent $j_t$ arrives in round $t$, we set $(y_t)_j = 1$ if $j = j_t$ and $(y_t)_j = 0$ otherwise. Observe that for any $\pi \in \Pi$, $U(\pi, j_t) = \langle z^\pi, y_t \rangle$. Hence, our pseudo-regret $R_T$ is equal to the adversarial linear bandit pseudo-regret:

$$R_T = \max_{z \in \mathcal{Z}} \left[ \sum_{t=1}^{T} \langle z, y_t \rangle - \sum_{t=1}^{T} \langle z_t, y_t \rangle \right].$$

Using the EXP3 algorithm for linear bandits, we obtain a regret upper bound of $\mathcal{O}\sqrt{KT \log(N)}$ for our regret minimization problem as $|\mathcal{Z}| = \mathcal{O}(KN)$. However, this upper bound can be further improved as follows. Let $\mathcal{R} = \{e_j\}_{j \in [K]}$ be the set of all possible reward vectors. Define $\mathcal{Z}_0$ as the smallest subset of $\mathcal{Z}$ such that

$$\max_{z \in \mathcal{Z}} \min_{z' \in \mathcal{Z}_0} \max_{y \in \mathcal{R}} |\langle z - z', y \rangle| \leq \frac{1}{T}.$$

By the above inequality, it suffices to focus on $\mathcal{Z}_0$ for our regret minimization problem. It can be shown that $|\mathcal{Z}_0| \leq \min\{2KN + N + 1, (6KT)^K\}$. Using the EXP3 algorithm for linear bandits on the reduced arm set $\mathcal{Z}_0$, we achieve a regret upper bound of $\mathcal{O}\left( \sqrt{KT \log(N)}, \min\{K\sqrt{T \log(KT)}\} \right)$.

# E. Smooth Demand with Unknown Agent Types

### E.1. Proof of Theorem 4.1

*Proof.* We consider the instance were the principal reward of first $N - 1$ products is equal to 1, and the last product is 0. We partition $[0, \frac{1}{2}]$ incentive for each arm by $\lfloor \epsilon \rfloor$ equal parts, and we propose a class of agent types $\{A^{i,j}\}_{i \in [N-1], j \in [K]}$, were given incentive $\pi$, the agent's selection probabilities is defined by Eq. 1-2. The bonus function is defined as $\mathbf{B}(x) := \frac{L-1}{4} \min(x, \epsilon - x)$. Consequently, the expected principal reward when agent type $A^{i,j}$ arrives and incentive $\pi$ is offered with nonzero element to arm $i$ would be

$$\mathbb{E}[r(\pi)] = \sum_{l=1}^{N-1} (1 - \pi_l) \Pr[a(\pi, (i, j))] = \begin{cases} \frac{N-1}{16N} + (1 - \pi_i)\mathbf{B}(\pi_i - j\epsilon) & \text{if } \pi_i \in [j\epsilon, (j+1)\epsilon] \\ \frac{N-1}{16N} & \text{if } \pi \in [0, \frac{1}{2}]^N \\ \leq \frac{N-1}{16N} & \text{otherwise.} \end{cases}$$

**Lemma E.1.** *For all agent types $(i, j)$, Assumption 1.1 holds.*

For all $\pi, \pi'$ where $\|\pi - \pi'\|_\infty = \Delta$, let $l(\pi)$ and $l(\pi')$ be the indices were incentive is nonzero respectively (if $\pi = \mathbf{0}$, then $l(\pi) = N + 1$)

$$
\begin{aligned}
\sum_{l=1}^{N} \big| \Pr[a(\pi, (i,j))) = l] - \Pr[a(\pi', (i,j)) = l] \big| &= \big| \Pr[a(\pi, (i,j))) = l(\pi)] - \Pr[a(\pi', (i,j)) = l(\pi)] \big| \\
&\quad + \big| \Pr[a(\pi, (i,j))) = l(\pi')] - \Pr[a(\pi', (i,j)) = l(\pi')] \big| \\
&\quad + \big| \Pr[a(\pi, (i,j))) = N] - \Pr[a(\pi', (i,j)) = N] \big| \\
&= \Pr[a(\pi, (i,j))) = l(\pi)] - \frac{1}{16N} \\
&\quad + \Pr[a(\pi', (i,j))) = l(\pi')] - \frac{1}{16N} \\
&\quad + \big| \Pr[a(\pi, (i,j))) = l(\pi)] - \Pr[a(\pi', (i,j)) = l(\pi')] \big| \\
&\leq 4 \max_{x \leq \frac{1}{2}} \big| \frac{d}{dx} \frac{1}{16N(1-x)} \big| \Delta + 4\mathbf{B}(\min(\epsilon, \Delta)) \\
&\qquad\qquad\qquad\qquad\qquad\qquad\qquad\qquad \text{(Mean value theorem)} \\
&\leq \Delta + (L-1)\Delta = L\Delta
\end{aligned}
$$

Therefore, if an algorithm can't play the good interval incentive of each agent type consistently, the expected regret would be $\frac{1}{8}(L-1)\epsilon T$, setting $\epsilon = (L-1)^{-2/3}N^{1/3}T^{-1/3}$, the regret would be $\frac{1}{8}(L-1)^{1/3}N^{1/3}T^{2/3}$. For $i \in [N], j \in \lceil \epsilon \rceil$, let $\mathcal{D}_{i,j} = \{\pi \in [0,1]^N | \pi_i \in [j\epsilon, (j+1)\epsilon), \pi_{i'} = 0 \quad \forall i' \neq i\}$, then $\pi_{i,j}^* \in \mathcal{D}_{i,j}$, for all $(i,j) \in J$. So for any algorithm *Alg* on the problem instance $(i, j) \in J$, we reduce it to an algorithm *Alg'* in stochastic MAB setting instance with $\mathcal{A} = \{(i', j') | i' \in [N], j' \in \lceil \epsilon^{-1} \rceil\}$ as the arm set. If algorithm *Alg* offers incentive $\pi \in \mathcal{D}_{i',j'}$ to agent, and receives reward $r(\pi)$; then *Alg'* selects arm $(i', j')$ and receives reward $\max_{\pi \in \mathcal{D}_{(i',j')}} \mathbb{E}[r(\pi)] + \eta_t$, where $\eta_t \sim \mathcal{N}(0, 1)$. So the expected regret of algorithm Alg is lower-bounded by the expected regret of Alg'. For instance $(i, j) \in J$, the expected reward of the reduced MAB problem is bounded by

$$
r'_{(i,j)}(i', j') = \begin{cases} \geq \frac{N-1}{16N} + \frac{(L-1)\epsilon}{16} & \text{if } (i', j') = (i, j) \\ \leq \frac{N-1}{16N} & \text{otherwise.} \end{cases}
$$

Using instance-dependent lower bound for MAB with Gaussian noise ((Lattimore & Szepesvári, 2020), Section 16.2), we have

$$
\inf_{\text{Alg}} \sup_{j \in J} R_T \geq \inf_{\text{Alg'}} \sup_{j \in J} R'_T = \Omega(\sum \Delta_{(i,j)}^{-1}) = \Omega((L-1)^{1/3}N^{1/3}T^{1/3})
$$

$\square$

## E.2. Equivalence to greedy model with Gaussian noise

**Lemma E.2.** *If for an agent of type $j$ arriving at time $t$, their preference vector is $\mu^j + \eta^t$, where $\mu^j$ is the expected preference vector if agents of type $j$, and $\eta^t \sim \mathcal{N}(\mathbf{0}, I)$ is independent noise of the agent, then for Greedy choice model Assumption 1.1 holds.*

*Proof.*

$$
\begin{aligned}
\Pr[a(\pi_t, j_t) = i^*] &= \Pr[i^* \in \arg\max_{i \in [N]} \left\{ \mu_i^{j_t} + \pi_{t,i} + \eta_i^t \right\}] \\
&= \Pr[\eta_{i^*}^t + c_{i^*} \geq \max_{i \neq i^* \in [N]} \eta_i^t + c_i] \qquad\qquad \text{(define } c_i := \mu_i^{j_t} + \pi_t, i)
\end{aligned}
$$

$$= \int \Pr[\eta_{i^*}^t + c_{i^*} = y] \Pr[\max_{i \neq i^* \in [N]} \eta_i^t + c_i \leq y] dy$$

$$= \int \phi(y - c_{i^*}) \Big( \prod_{i \neq i^*}^{N} \Phi(y - c_i) \Big) dy$$

where $\phi$ and $\Phi$ are PDF and CDF of standard normal distribution respectively. Since $\Pr[a(\pi_t, j_t) = i^*]$ is everywhere differentiable, and $\nabla_\pi \Pr[a(\pi_t, j_t) = i^*]$ is bounded over $\pi \in [0, 1]^N$, using mean-value theorem, it's also Lipschitz. $\quad\square$

