# OpenReview forum: "Learning to Incentivize in Repeated Principal-Agent Problems with Adversarial Agent Arrivals"
_ICML.cc/2025/Conference — ICML 2025 poster_

### Official Review · Reviewer_wWxu · 2025-03-10

**Overall Recommendation:** 4

**Summary:**

This work explores sequential incentive design in a repeated principal-agent problem with adversarially ordered agents. The principal faces $K \geq 2$ agent types (unknown) and selects incentives for one of $N$ arms to influence agent decisions, which are made based on both intrinsic utility and offered incentives. The goal is to minimize regret relative to the optimal ex-post incentive strategy. The contributions of this paper are:

1. Demonstrate that algorithms lacking prior knowledge of agent behaviors (e.g., best-response functions) inevitably incur linear regret in greedy-response, single-arm incentive settings.

2. Propose a reduction-based approach for adversarial linear bandits by discretizing the continuous incentive space. Achieved an upper bound of O(min{ $ \sqrt{KT log(KN)}, K\sqrt{T}$ }) for greedy-response agents with known types.

3. Introduce a novel polytope discretization for large incentive spaces, enabling $O\left(K\sqrt{T}\right)$ regret when incentivizing multiple arms.

4.  Design an algorithm achieving $\tilde{O}\left(L^{1/3}N^{1/3}T^{2/3}\right)$ regret for Lipschitz-smooth agent responses.

This study pioneers the analysis of adaptive incentive mechanisms against adversarial agent arrivals, establishing tight regret bounds under assumptions and addressing exploration-exploitation trade-offs in dynamic incentive allocation.

**Claims And Evidence:**

In lines 192-196, the authors claim that to achieve sub-linear regret, the method must exactly learn $\Delta$. In this paper, $\Delta \in [0.7, 0.71]$. Does this imply that achieving sub-linear regret is difficult only when $\Delta$ is relatively small?

**Essential References Not Discussed:**

None.

**Experimental Designs Or Analyses:**

There are no experiments.

**Methods And Evaluation Criteria:**

The method makes sense, and there are no evaluation datasets.

**Other Comments Or Suggestions:**

1. In line 149, " $L \geq$ " is incomplete.

**Other Strengths And Weaknesses:**

**Strengths**

1. This paper provides a detailed discussion on the upper and lower bounds of the repeated principal-agent problem under different agent behaviors and incentive strategies.

2. This article has a clear logical structure and rigorous proofs for the theorems.

3. This paper innovatively defines a polytope to address the issue of an excessively large incentive space, using the extreme points of the polytope to determine the optimal incentive.

**Questions For Authors:**

In Section 2.2, the arms are mapped to {0, 1}, and the rewards are approximately set to 0.5. Additionally, in Section 3.1, the assumptions regarding tie-breaking have been simplified. If the discretization were to be closer to the real setting and more complex, would it affect the derivation of the upper and lower bounds?

**Relation To Broader Scientific Literature:**

This paper focuses on repeated principal-multi-agent problems, which have been extensively studied. The difference lies in the fact that this paper emphasizes agents arriving in an adversarial order, whereas previous work assumes that agents arrive in a fixed distributional order.

**Theoretical Claims:**

I have checked the correctness of any proofs for theoretical claims.

---

> ### Author Rebuttal · Authors · 2025-04-01
>
> We thank the reviewer for their review. We now address their concerns.
>
> **Typo regarding $L$:** Thanks for pointing out this typo. This should be $L \ge 1$.
>
> **Regarding the parameter $\Delta$ in our linear-regret lower bound:** In the principal-agent problem, the incentives provided to any arm lie within the range $[0,1]$. In our lower bound instance, the optimal incentive to provide to arm $1$ is $\Delta$; any deviation above or below this value results in linear regret for the algorithm. We select $\Delta \in [0.7, 0.71]$, which is relatively large on the $[0,1]$ scale.
>
> **Response to reviewer's question:** Our upper bound applies to all scenarios; the tie-breaking assumption is made only for simplicity of presentation. We have addressed all possible tie-breaking scenarios in the appendix. In contrast, our lower bound is a worst-case bound. Thus, analogous to multi-armed bandit problems, establishing instance-dependent upper and lower bounds in this setting would be an interesting direction for future research.

---

### Official Review · Reviewer_tqFh · 2025-03-11

**Overall Recommendation:** 4

**Summary:**

Building upon the literature on repeated principal-agent games, this paper explores a setup where a principal recommends an action from a bandit instance to an agent and offers a payment so the agent is incentivized to follow the recommendation. Two cases are studied: first when the agent greedily chooses her utility and second, when the agent chooses her action with a smooth choice model. The difference with the state-of-the-art here is that the agent's type varies over type and is unobserved to the principal. To stick with the contract theory terminology, it is a setup with observed action and unknown type (i.e. adverse selection). The adversarial arrival of agents of different types makes it necessary to introduce a new definition of regret and to use a discretization of the incentives' space before linking the problem to adversarial bandits.

**Claims And Evidence:**

Yes.

**Essential References Not Discussed:**

NA.

**Experimental Designs Or Analyses:**

There are no experiments in this paper.

**Methods And Evaluation Criteria:**

There are no experimental evaluation criteria beyond a theoretical regret bound. It is not an issue to avoid experimental evaluation to me, since the paper is mostly about theoretical issues (as is the literature in this field).

**Other Comments Or Suggestions:**

I would have appreciated a clear statement of the proposed algorithms. I believe that it would improve readability.

**Other Strengths And Weaknesses:**

I really appreciate the setup: it seems to interesting to me to account for adversarial agents' types. Also, I enjoy the technical approach which consists in first discretizing the incentives' space and then running an adversarial bandit algorithm. The work is nicely supported by a lot of lower bounds provided.

I appreciate having an outline of the important proofs in the main text and the full proof in the appendix.

From my reading, the regret upper bounds in page 8 for Instance-dependent Algorithm for Single-Arm/General Incentives are stated without a theorem nor a clear proof. I think that the paper would benefit from a clear presentation of the algorithm used in that setup (including the application of the Zooming algorithm), a formal theorem as well as proofs. The employed tools seem very interesting but the way they are presented in the current version is definitely too brief.

**Questions For Authors:**

Could you discuss a bit more the relation between the technical issues of your approach as compared to Zhu et al., The sample complexity of online contract design? Especially the way of discretizing the contract/incentives space.

Is the definition of regret that you give common in the unknown opponent principal-agent literature? Or in contract design more generally.

Do you believe that tackling the extension where the principal only observes noisy rewards is doable or close in the analysis?

**Relation To Broader Scientific Literature:**

The paper provides a consistent literature review in two areas: principal-agents problems (an economics' field) and online learning. Relying on recent works that consider repeated principal-agent problems (typically Dogan et al. Repeated principal-agent games with unobserved agent rewards and perfect-knowledge agents, Scheid et al., Incentivized learning in principal-agent bandit games, and then Ben-Porat et al., Principal-Agent Reward Shaping in MDPs which considers a MDP setup, among other extensions), the paper introduces a new complexity to the problem which is the adversarial agents arrivals (with different types).

To tackle it, the paper introduces a regret definition very close to the one defined in Zhu et al., The sample complexity of online contract design (this paper studies a repeated contract design setting with unobserved action and unknown types) and uses a discretization of the incentives' space a bit similar. While the discussion on the relation with the [repeated] principal-agent literature seem consistent to me, a  discussion with the latter reference might be interesting.

**Theoretical Claims:**

All the claims of the paper are supported by clear proofs.

---

> ### Author Rebuttal · Authors · 2025-04-01
>
> We thank the reviewer for their review. We now address their concerns.
>
> **On instance-dependent Algorithm for the smooth setting:**
>
> For the smooth setting, after establishing the minimax regret bounds, we directly apply the Zooming algorithm from [1] in Section 4.3 of our paper. The instance-dependent regret bound then follows from Theorem 3.1 in [1], which characterizes the regret in terms of the zooming dimension. We agree that a clearer presentation would benefit the reader; thus, if space permits, we will include a pseudocode outline of the Zooming algorithm along with a precise statement of the associated regret bound, citing Theorem 3.1 of [1] for completeness.
>
> [1] Podimata, C. and Slivkins, A. Adaptive Discretization for Adversarial Lipschitz Bandits. In *Proceedings of the Thirty-Fourth Conference on Learning Theory*, vol. 134, pp. 3788–3805, 2021.
>
> **Comparison with Zhu et al.**
>
> Since both papers consider settings where the agent best responds to the incentive, a key technical challenge in both is that the principal’s utility function is not Lipschitz continuous with respect to the incentive. That is, small changes in incentives can lead to abrupt shifts in agent behavior. However, the approaches for handling this non-smoothness differ significantly. Zhu et al. (2022) identify directions in the incentive space along which the utility function is continuous and construct a discretization based on spherical codes to cover these directions.
>
> In contrast, our approach leverages structural knowledge of agent behavior: for any given incentive, we know exactly which arm each agent type will choose. This allows for a more tailored discretization strategy. For instance, in the single-arm incentive setting, we enumerate threshold points across agent types and prune the incentive set by retaining only the most rewarding vectors, making the discretization independent of $N$. In the general incentive setting, we again use the knowledge of agents’ best responses to characterize the incentive space as a polytope, and the algorithm only needs to consider the extreme points of that polytope.
>
> **On the Prevalence of the Regret Definition in the literature:**
>
> As we highlight in the paper, our work initiates a new problem setting that combines elements of the principal-agent framework with adversarially ordered arrivals. In this setting, the regret notion we adopt arises naturally, and closely related definitions have been studied in the context of Stackelberg games. For instance, the following works—one of which was also cited by Reviewer QVD9—explicitly consider this form of regret:
>
> *References*
> 1. Harris et al. Regret Minimization in Stackelberg Games with Side Information, NeurIPS 2024.
> 2. Balcan, Maria-Florina, et al. Commitment without regrets: Online learning in Stackelberg security games. Proceedings of the Sixteenth ACM Conference on Economics and Computation, 2015.
>
> Although this exact regret formulation may not have been explored in the contract design literature to our knowledge, we believe it is a natural and principled objective in our setting. As such, it may help inspire new directions in principal-agent and contract-design problems under adversarial settings.
>
> **On the case when the principal receives only noisy feedback:**
>
> Under stochastic bandit feedback, our upper-bound results for single-arm incentives under the greedy model extend naturally to the case where the principal’s rewards are unknown, as follows. We retain the current discretization of the incentive vector space and run Tsallis-INF over these discretized points, incurring a regret of $\sqrt{KNT}$. However, in contrast to the known principal-reward setting, the dependence on $N$ cannot be improved to achieve a regret bound of $\min\left(\sqrt{KT\log N}, K\sqrt{T}\right)$. This is because an $N$-armed stochastic multi-armed bandit problem can easily be reduced to our principal-agent problem, and it is known that an $N$-armed stochastic multi-armed bandit problem has a regret lower bound of $\Omega\left(\sqrt{NT}\right)$.

---

> > ### Comment · Reviewer_tqFh · 2025-04-05
> >
> > I thank the authors for their detailed and interesting answers to my questions and I thus increase my grade.

---

### Official Review · Reviewer_QVD9 · 2025-03-14

**Overall Recommendation:** 4

**Summary:**

This paper studies a repeated principal-agent game, where the principal delegates their action to $K$ agents, each choosing from $N$ actions, with agent types ($i_t \in [K]$) assigned adversarially in each round. First, they show that achieving no-regret requires the principal to have prior knowledge of the agents' behavior—specifically, access to their best response functions—when agents act greedily (maximizing their utility in each round). Under this greedy action model, the paper presents an algorithm with $O(\min(\sqrt{KT\log N}, K\sqrt{T}))$ regret (in the single-arm incentive case) and proves a matching lower bound up to an $O(\log K)$ factor. They then extend their results to a smoothed action model, where agents choose arms probabilistically based on an unknown distribution that varies smoothly with the incentive vector (the principal's payout vector for each arm). In this smoothed model and the single-arm incentive case, they provide a no-regret algorithm and a matching lower bound.

**Claims And Evidence:**

This paper is mainly theoretical and its claims are supported by proofs.

**Essential References Not Discussed:**

There is a missing prior work, Harris et al. (2024), which studies a very similar problem setting. See the weaknesses section for details.

**Experimental Designs Or Analyses:**

N/A

**Methods And Evaluation Criteria:**

N/A

**Other Comments Or Suggestions:**

- Table 1 could include pointers to the corresponding theorem numbers to improve readability.
- A proof sketch or illustration of the main idea or strategy in the main text would be very helpful.
- Formally stating the regret upper bound for the smoothed case as a theorem would enhance the paper's coherence, readability, and navigability.

**Other Strengths And Weaknesses:**

**Strengths:**
- This paper studies a strong setting in the repeated principal-agent game where the agent's type is chosen adversarially.
- It provides a solid negative result, motivating the necessity for the principal to know each agent's best response rule.
- For their algorithms, they establish nearly matching lower bounds.

**Weaknesses:**
- (Minor weakness) Lack of concrete applications for the studied problem setting.
- The setting of this paper appears to be subsumed by that of Harris et al. (2024), which studies a general principal-agent (Stackelberg) game with side information, adversarial agent-type arrivals, and bandit feedback (where the principal observes only the agent's chosen action). While Harris et al. achieve a regret bound of $O(T^{2/3})$ in this setting, worse than that of this work (Note that, a concurrent result by Balcan et al. (2025) improves this to $O(T^{1/2})$), as their setting includes side information, it inherently considers a stronger regret benchmark. Given the similarities in problem formulation, a comprehensive comparison between this work and Harris et al. seems necessary.

**Reference**
- Harris et al. Regret Minimization in Stackelberg Games with Side Information, *NeurIPS* 2024.
- Balcan et al. Nearly-Optimal Bandit Learning in Stackelberg Games with Side Information *arXiv:2502.00204* 2025.

**Questions For Authors:**

Is it possible to extend the result to the case where the principal's rewards are unknown, but the principal receives only bandit feedback?

**Relation To Broader Scientific Literature:**

This paper contributes to the literature on learning in repeated principal-agent games, particularly in the setting where agent types arrive adversarially, providing near-optimal algorithms for this scenario. Prior works have primarily considered settings where the principal repeatedly contracts with a fixed but unknown agent type (Scheid et al., 2024b; Dogan et al., 2023a,b) or where the agent's type is drawn stochastically from a fixed distribution (Ho et al., 2014; Gayle & Miller, 2015). While some prior settings have advantages—e.g., Scheid et al. study the case where the principal's reward vector is drawn from an unknown distribution, whereas this work assumes a known reward vector—this paper's adversarial arrival model is relatively novel (but not completely new, see the section below) and warrants further study.

**Theoretical Claims:**

I found no specific issues with their proofs, particularly the main upper bounds in Theorem 3.1, Theorem 3.2, and the unnamed no-regret result in Section 4.1.

---

> ### Author Rebuttal · Authors · 2025-04-01
>
> We thank the reviewer for their review. We now address their concerns.
>
>  **On Presentation suggestions:** We thank the reviewer for the helpful suggestions regarding presentation. We will carefully incorporate them to improve clarity and readability in the next version.
>
>
> **Comparison with Harris et al.**
>
> We thank the reviewer for referring us to the recent NeurIPS paper by Harris et al. (2024) and the follow up concurrent arXiv preprint by Balcan et al. (2025). While these works share similarities with our framework—such as a principal-agent setup (which corresponds to the leader-follower formulation in their paper), with unknown agent types appearing each round—their settings differ in several important ways that prevent them from subsuming our problem:
>
> 1. *Action Space*: In our setting, the principal chooses an incentive vector from the hypercube $[0,1]^N$. In contrast, in the work by Harris et al., the leader selects a mixed strategy from a probability simplex defined over a finite set of actions $\mathcal{A}$.
>
>
> 2. *Agent/Follower Rewards*: In our model, if agent $j$ selects arm $i$, they obtain a reward of $\mu_i^j + \pi_i$, where $\pi$ is the incentive vector chosen by the principal. Conversely, in the leader-follower mode of Harris et al., the follower's reward for choosing action $a_f$ is $\sum_{a_\ell \in \mathcal{A}} x[a_\ell] u_j(z, a_\ell, a_f)$, where $x$ is the mixed strategy chosen by the leader and $z$ represents contextual information.
>
> 3. *Principal/Leader Rewards*: In our setting, if arm $i$ is chosen, the principal’s reward is $v_i - \pi_i$. On the other hand, in Harris et al., if the follower chooses action $a_f$, the leader’s reward is $\sum_{a_\ell \in \mathcal{A}} x[a_\ell] u(z, a_\ell, a_f)$, where $x$ is the mixed strategy chosen by the leader and $z$ is contextual information.
>
> These distinctions—particularly in the action spaces and the reward structures—mean that the Stackelberg game setting of Harris et al. (2024) does not subsume our principal-agent formulation. Nonetheless, we will clarify this comparison in our paper and emphasize that extending our principal-agent framework to a contextual setting, akin to theirs, would be a promising direction for future work.
>
> **On concrete applications of our problem setting:** While we briefly discussed motivating examples in the introduction, we are happy to expand on them here to further clarify how the studied problems apply to real-world scenarios.
>
> *Adversarial arrival of agents.* In practice, agent arrivals often deviate from fixed stochastic patterns due to factors like non-stationarity, strategic behavior, or external influences. For instance, in online shopping, discount ads are shown to all users, but only some choose to act—often in response to timing, personal context, or even social trends. The sequence of users who respond may not follow any stable distribution. Modeling arrivals adversarially allows us to account for such unpredictable and potentially strategic participation without assuming a specific arriving distribution.
>
> *Agent response models.* For additional motivating examples of the best response and smooth response models, please refer to our detailed response to Reviewer LMoE.
>
> **On the case when the principal receives only bandit feedback:**
>
> Under stochastic bandit feedback, our upper-bound results for single-arm incentives under the greedy model extend naturally to the case where the principal’s rewards are unknown, as follows. We retain the current discretization of the incentive vector space and run Tsallis-INF over these discretized points, incurring a regret of $\sqrt{KNT}$. However, in contrast to the known principal-reward setting, the dependence on $N$ cannot be improved to achieve a regret bound of $\min\left(\sqrt{KT\log N}, K\sqrt{T}\right)$. This is because an $N$-armed stochastic multi-armed bandit problem can easily be reduced to our principal-agent problem, and it is known that an $N$-armed stochastic multi-armed bandit problem has a regret lower bound of $\Omega\left(\sqrt{NT}\right)$.

---

> > ### Comment · Reviewer_QVD9 · 2025-04-09
> >
> > Thank you for your response, which clarifies the relationship between this paper and the prior/concurrent works. I recommend including brief pointers to the relevant references, along the lines of what was outlined in the rebuttal, in the revision. I have raised my score based on your response.

---

### Official Review · Reviewer_LMoE · 2025-03-19

**Overall Recommendation:** 3

**Summary:**

The paper introduces a repeated principal-agent setting where agents arrive in an adversarial fashion. The principal interacts with agents of unknown types by strategically offering incentives to influence their decisions. The paper proposes algorithms with sublinear regret bounds under two key settings: (1) when the principal knows the best response of each agent type and (2) when agent decisions vary smoothly with incentives. The authors also present matching lower bounds for both settings and extend the results to cases where the principal can incentivize multiple arms simultaneously.

## update after rebuttal

I thank authors for their detailed responses. The paper made reasonable amount of advancement in the existing literature, but I still do not see any noteworthy technical breakthrough here.  Hence, I am keeping my original score.

**Claims And Evidence:**

yes

**Essential References Not Discussed:**

n/a

**Experimental Designs Or Analyses:**

n/a

**Methods And Evaluation Criteria:**

yes

**Other Comments Or Suggestions:**

n/a

**Other Strengths And Weaknesses:**

The paper begins by formalizing the hard cases for the repeated principal-agent problems in general and then introduces natural conditions where it is possible to design no-regret learning algorithm. Overall, it is a nice complement to the existing literature on repeated principal-agent problem. The paper is overall well-written. That said, I do not see any technical breakthrough here, as the designs and analysis of these algorithms are well-expected.

**Questions For Authors:**

Can you provide some real-world motivation on the two models you consider?

**Relation To Broader Scientific Literature:**

The adversarial setup is a meaningful addition to the existing literature in repeated principal-agent problem.

**Theoretical Claims:**

I only read through the proof sketch

---

> ### Author Rebuttal · Authors · 2025-04-01
>
> We thank the reviewer for their review. We now address their concerns.
>
> **On technical breakthrough:** While we respect the reviewer’s opinion, we believe our work includes notable technical breakthroughs. We develop novel lower bound techniques tailored to the greedy model setting—methods that, to the best of our knowledge, are new and have the potential to extend to other related problems like regret minimization in linear bandits and stackelberg games. Additionally, we develop a new discretization approach and provide a reduction to linear bandits, which together yield near-optimal regret bounds. To the best of our knowledge, this reduction strategy is novel and has not been explored in prior work.
>
> **Real world motivation of the two models:**
>
> While we briefly discussed motivating examples in the introduction, we are happy to expand on them here to further clarify the relevance of the two models to real-world scenarios.
>
> *Best response model.* A natural example arises in online shopping, where customers make purchase decisions based on visible discounts (i.e., incentives). Customers often wait until a discount reaches a historical low or crosses a personal threshold before making a purchase (Gunadi & Evangelidis, 2022). Similarly, in online labor markets, crowdworkers frequently accept tasks only if the offered payment exceeds a minimum expected amount (Horton & Chitoni, 2010). These behaviors reflect a threshold-based decision process, motivating our use of the best response model. In this model, the agent plays an arm only when the incentive on that arm exceeds a predefined threshold.
>
> *Smooth response model.* In contrast, consider routine purchases such as daily necessities. Here, small changes in discounts do not lead to abrupt changes in behavior but instead gradually affect the probability of purchase (Bijmolt et al., 2005). A similar pattern is observed in ad click-through behavior, where users’ likelihood of clicking on an ad increases with the attractiveness of the offer—such as a better discount or a more personalized promotion—but does so in a smooth, probabilistic manner rather than through sharp thresholds (Bleier & Eisenbeiss, 2015). These scenarios motivate the need for a more flexible response model in which decisions vary smoothly with the incentive.
>
> *References*:
>
> Gunadi, M. P., & Evangelidis, I. (2022). The impact of historical price information on purchase deferral. Journal of Marketing Research, 59(3), 623–640.
>
> Horton, J. J., & Chilton, L. B. (2010). The labor economics of paid crowdsourcing. In Proceedings of the 11th ACM conference on Electronic commerce (pp. 209-218).
>
> Bijmolt, T. H., Van Heerde, H. J., & Pieters, R. G. (2005). New empirical generalizations on the determinants of price elasticity. Journal of marketing research, 42(2), 141-156.
>
> Bleier, A., & Eisenbeiss, M. (2015). Personalized online advertising effectiveness: The interplay of what, when, and where. Marketing Science, 34(5), 669-688.

---

### Decision · Program_Chairs · 2025-05-01

**Decision:**

Accept (poster)

**Comment:**

All the Reviewers agreed that the paper clearly passes the bar for acceptance at ICML. Thus, I will propose acceptance.